

# Ensemble Forecasts of Air Quality in Eastern China
## Part 2. Evaluation of the MarcoPolo-Panda Prediction System, Version 1.

Anna Katinka Petersen[1], Guy P Brasseur[1,2], Idir Bouarar[1], Johannes Flemming[3], Michael Gauss[4], Fei Jiang[5], Rostislav Kouznetsov[6], Richard Kranenburg[7], Bas Mijling[8], Vincent-Henri Peuch[3], Matthieu Pommier[4], Arjo Segers[7], Mikhail Sofiev[6], Renske Timmermans[7], Ronald van der A[8,9], Stacy Walters[2], Ying Xie[10], Jianming Xu[10], Guangqiang Zhou[10]

[1] Max Planck Institute for Meteorology, Hamburg, Germany
[2] National Center for Atmospheric Research, Boulder, CO, USA
[3] European Centre for Middle Range Weather Forecasts, Reading, UK.
[4] Norwegian Meteorological Institute, Oslo, Norway
[5] Nanjing University, Nanjing, China
[6] Finnish Meteorological Institute, Helsinki, Finland.
[7] TNO, Utrecht, The Netherlands
[8] Royal Netherlands Meteorological Institute (KNMI), De Bilt, The Netherlands
[9] Nanjing University of Information Science and Technology, Nanjing, China
[10] Shanghai Meteorological Service, Shanghai, China

## Abstract:

An operational multi-model forecasting system for air quality has been developed to provide air quality services for urban areas of China. The initial forecasting system included seven state-of-the-art computational models developed and executed in Europe and China (CHIMERE, IFS, EMEP MSC-W, WRF-Chem-MPIM, WRF-Chem-SMS, LOTOS-EUROS and SILAMtest). Several other models joined the prediction system recently, but are not considered in the present analysis. In addition to the individual models, a simple multi-model ensemble was constructed by deriving statistical quantities such as the median and the mean of the predicted concentrations.

The prediction system provides daily forecasts and observational data of surface ozone, nitrogen dioxides and particulate matter for the 37 largest urban agglomerations in China (population higher than 3 million in 2010). These individual forecasts as well as the multi-model ensemble predictions for the next 72 hours are displayed as hourly outputs on a publicly accessible web site (www.marcopolo-panda.eu).

In this paper, the performance of the predictions system (individual models and the multi-model ensemble) for the first operational year (April 2016 until June 2017) has been analysed through statistical indicators using the surface observational data reported at Chinese national monitoring stations. This evaluation aims to investigate a) the seasonal behavior, b) the geographical distribution and c) diurnal variations of the ensemble and model skills. Statistical indicators show that the ensemble product usually provides the best performance compared to the individual model forecasts. The ensemble product is robust even if occasionally some individual model results are missing.

Overall and in spite of some discrepancies, the air quality forecasting system is well suited for the prediction of air pollution events and has the ability to provide alert warning (binary prediction) of air pollution events if bias corrections are applied to improve the ozone predictions.



## 1. Introduction

With the rapid development of its economy, China has been experiencing repeated intense air pollution episodes (e.g. *Guo et al., 2014, Huang et al., 2014, Wang et al., 2014*) with a wide range of health effects (*Kampa and Castanas 2008; Wu et al., 2012; Hamra et al. 2015; Boynard et al., 2014; WHO, 2018*) and serious consequences on ecosystems (*Fowler et al., 2008, Ashmore, 2005; Leisner et al., 2012; Sinha et al., 2015*) and on climate (*Sitch et al.* 2007; *Brasseur et al., 1999; Akimoto, 2003*). High concentrations of particulate matter often cover a large area of eastern China during winter when air remains stagnant for several days and chemical compounds emitted by power plants, industrial complexes, traffic and domestic infrastructures remain trapped near the surface (e.g. *Wang et al., 2014; Zhao et al., 2013*). During summer, photochemical processes convert nitrogen oxides ($NO_X$) and volatile organic compounds (VOCs) into tropospheric ozone ($O_3$) (e.g. *Xu et al., 2008, Sun et al., 2016*).

Long-term solutions to mitigate air pollution require a fundamental transformation of the energy system, which may require decades to be fully implemented. Short-term actions to avoid severe air pollution episodes, however, can be put in place immediately if such episodes can be reliably predicted a few days prior to their occurrence. Comprehensive air quality models that capture meteorological, chemical and physical processes in the troposphere and predict the fate of air pollutants are key tools to forecast the likelihood of air pollution episodes and hence to inform the authorities.

Within the EU projects MarcoPolo and Panda, that include European as well as Chinese partner organizations, an operational multi-model forecasting system for air quality including a number of different chemical transport models has been developed, and is providing daily forecasts of ozone, nitrogen oxides, and particulate matter for the 37 largest urban areas of China (population higher than 3 million in 2010). These individual forecasts as well as the mean and median concentrations for the next 3 days are posted on a dedicated website (www.marcopolo-panda.eu/forecast) together with the hourly observational data from local measurements reported by the Chinese monitoring network of the China National Environmental Monitoring Centre (CNEMC) (data available at www.pm25.in). This operational air quality analysis and forecasting system is presented in detail in a companion paper (*Brasseur et al, 2018*), where the individual models contributing to the MarcoPolo-Panda prediction system are described, and details about the individual models and their individual settings are provided. Information about selected parametrization options for the physical processes, including boundary layer, radiation, convection and surface processes, and about the emissions adopted in MarcoPolo-Panda prediction system are also provided.

In the present study, we evaluate the prediction system of the MarcoPolo and Panda projects that have been in operation for more than one year. We concentrate on the period April 2016 to June 2017 and analyse the model forecasts (7 individual models and the ensemble median) and observational data for 34 cities (covered by most of the models, depending on the extent of the domains, for two models only 31 and 32 cities).

We evaluate the performance of the individual models involved in the present study, and to examine the performance of the overall forecasting system by comparing the predicted surface concentrations to values reported by the Chinese air pollution monitoring network. Section 2 of the paper provides a brief description of the forecasting system, while Section 3 investigates the performance of the system using different statistical indicators including the mean bias (BIAS), the root mean square error (RMSE), the modified normalised bias (MNBIAS), the fractional gross error (FGE) and the correlation coefficient. We derive in particular (a) statistical indicators for each

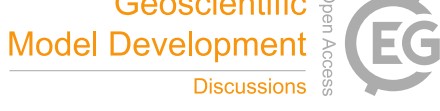



model over the time of the year (on a monthly basis) in order to analyse seasonal characteristics, (b)
the geographical distribution of the statistical indicators for the ensemble median in order to derive
regional characteristics and issues, (c) the statistical indicators of all models and of the ensemble
median over the time of the day (considering all model-observation pairs of all cities and for the
whole time period) and for a specific city (Beijing) together with the diurnal variation of the
pollutants during the whole time period. In Section 4, we assess the impacts of missing forecasts
from one or more models on the production of the ensemble. As the prediction system intends to
provide warning of air pollution episodes to the general public, the system performance has been
evaluated regarding its ability to predict the exceedence of air quality thresholds (binary prediction
of pollution events). This analysis is presented in Section 5. We show that the application of bias
correction to the models improves the forecasting skills of binary ozone predictions. We conclude
with a summary and outlook in Section 6.

## 2. Description of the Analysis and Forecasting System
Within the EU projects MarcoPolo and Panda, a number of chemistry transport models have been
applied to provide daily air quality forecasts for a selection of 37 large Chinese agglomerations
(population over 3 million, 2010 census). Initially, seven models, CHIMERE (Royal Netherlands
Meteorological Institute (KNMI)), IFS (European Centre for Medium Range Weather Forecast
(ECMWF)), WRF-chem-SMS (Shanghai Meteorological Service (SMS)), SILAMtest (Finish
Meteorological Institute (FMI)), WRF-chem-MPIM (Max Planck Institute for Meteorology
(MPIM) in Hamburg), EMEP MSC-W (hereafter referred to as 'EMEP', Norwegian Meteorological
Institute (MET Norway)) and LOTOS-EUROS (The Netherlands Organisation for Applied
Scientific Research (TNO)) were providing daily forecasts every day at 0:00 UTC for the next 72
hours (three days) for $NO_2$, $O_3$, PM10 and PM2.5 (see Figure 1). WRF-CMAQ and WRMS-
CMAQ, both used by Chinese institutions (Nanjing University and SMS), have joined recently the
prediction system, but are not considered in the present analysis.
We should note that the models considered in the present study may have significantly evolved
since the present analysis was performed. This is the case, for example, of the SILAM model
developed by the Finish Meteorological Institute, whose configuration was still in a test mode, and
is therefore referred to as SILAMtest.
The individual models are executed independently on the computing systems available in each
partner institution. The surface concentrations of the key chemical species are extracted locally
from the model outputs and forwarded to a central database operated by the Royal Netherlands
Meteorological Institute (KNMI).
Hourly predictions of surface concentrations (expressed in $\mu g/m^3$), are provided by the models as
grid values, which are bi-linearly interpolated to city center coordinates. The average for the data
provided by the urban network (usually around 5-12 stations), is posted together with the
corresponding standard deviation and the number of contributing stations.  In the present analysis,
we consider only the model simulations corresponding to 34 cities, since the cities of Ürümqi (most
western, only covered by three models), Changchun and Harbin (most northern cities), are located
outside of the domains covered by most individual models, which are indicated in the companion
paper (*Brasseur et al., 2018*).
In addition to the forecasts provided by the individual participating models, a multi-model ensemble
was constructed from which the median and the mean were derived. To process the ensemble





median, all seven individual models are first interpolated to a common horizontal grid. For each
grid point, the ensemble model is calculated as the median value of the individual model forecasts.
The median is relatively insensitive to outliers in the forecasts. The method is also less vulnerable to
occasionally missing data from individual models, as the minimum number of model results needed
to calculate a meaningful ensemble mean or median is almost always available. This will be
discussed in detail in Section 4. The multi-model approach also provides more accurate forecasts
and thus reduces the underlying uncertainties (as will be shown in the following section). More
advanced methods, e.g. based on individual model skills, are discussed in the literature (e.g.
*Galmarini et al, 2013*). They are significantly more costly from a computational point of view and
therefore not well suited for daily operations.

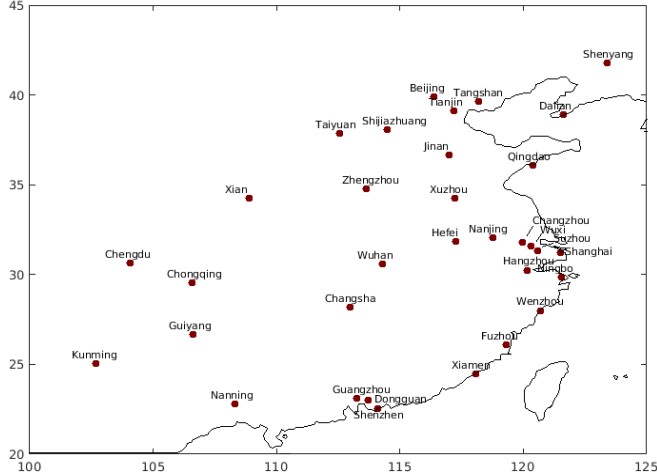


*Figure 1: Map of the 34 cities/urban clusters (population over 3 million (2010 census)) with available data (observational and model ensembles), used in this evaluation.*

## 3. Evaluation of the performance of the system

The evaluation of the performance of a forecasting system is a necessary step for assessing the
quality of the predictions and demonstrating its usefulness. It also provides important information
that can lead to the improvement of the forecasting system and to further model development. The
comparison between model output and in situ measurements is not straightforward because of the
different nature of the respective quantities: air quality models provide volume averaged quantities
over each model grid cell and time averages over the modeling time step. Observations are available
at fixed measurement sites and at a fixed time. Further, they are influenced by local processes that
are not necessarily well captured by relatively coarse models. Thus, the representativeness of the
observational site is not always guaranteed.
The MarcoPolo-Panda forecasting and analysis system uses the surface observations available at the
web site www.pm25.in for 37 Chinese cities. For a given city, the observational data considered for
the evaluation of the model consist of an average of the measurements made at the different stations
of the urban network, usually 5 – 12 stations, which are aggregated to one value for the whole city.
The model fields are bilinearly interpolated to the city center coordinates.
The mean bias






$$BIAS = \frac{1}{N}\sum_i(m_i - o_i),$$

where $m_i$ and $o_i$ are the model forecast value and the observation value, and $N$ the number of
model-observation pairs, the root mean square error

$$RMSE = \sqrt{\frac{1}{N}\sum_i(m_i - o_i)^2},$$

the modified normalized bias

$$MNBIAS = \frac{2}{N}\sum_i\frac{(m_i - o_i)}{(m_i + o_i)},$$

the fractional gross error

$$FGE = \frac{2}{N}\sum_i\left|\frac{m_i - o_i}{m_i + o_i}\right|$$

and the correlation coefficient between the model forecast and observed values

$$R = \frac{\frac{1}{N}\sum_i(m_i - \bar{m})\,(o_i - \bar{o})}{\sigma_m\sigma_o}$$

are used to measure the system performance. Here $\bar{m}$ and $\bar{o}$ are the mean values of the model
forecast and observed values, and $\sigma_m$ and $\sigma_o$ are the corresponding standard deviations.
The evaluation presented here aims to investigate a) the statistical indicators for each model over
the time of the year (on a monthly basis) so that the seasonal features can be characterized and
related issues of individual models can be identified (Section 3.1); b) the geographical distribution
of the statistical indicators of the ensemble median to highlight regional characteristics and related
issues (Section 3.2); c) statistical indicators of all models and the ensemble median over the time of
the day (considering all model-observation pairs of all cities and for the whole time period) and for
a specific city (Beijing) together with the diurnal variation of the pollution species over the whole
time period (Section 3.3).

## 3.1 Evaluation of the Seasonal Behavior of the Models
We start our evaluation of the multi-model prediction system by examining the seasonal behavior of
the predicted concentrations of key chemical species. The statistical indicators mentioned above
have been calculated separately for each month from April 2016 to June 2017 and for the entire
period during which the forecasting system was operational. Due to storage issues, only the
predictions for the first 24 hours (0-23h) were saved while the predictions from 24h-72h were not
retained and not analyzed in this work.



Figure 2 shows the RMSE, BIAS, MNBIAS and FGE of $NO_2$ (left panel) and $O_3$ (right panel) for
each of the seven individual models included in the system and for the model ensemble median, for
each individual month between April 2016 and June 2017. The same results are also provided for
the whole period ("all"). It can be seen, that there is a wide spread of the results produced by the
seven models.  The individual models are continuously improving during the first months because
many changes have been applied by the different modeling groups in order to improve their
individual predictions. In the case of $NO_2$, most individual models slightly overestimate the
concentrations compared to observations. In the EMEP model, it may be explained by the larger
nitric oxide emissions used in comparison with the other models (Brasseur et al., 2018).  This
results in a positive BIAS and MNBIAS for most models and the ensemble median. The RMSE of
the model ensemble is highest in July/August/September 2016 and remains relatively constant after
October 2016. It can be seen, that the median of the model ensemble has the lowest RMSE for $NO_2$,
the smallest BIAS and MNBIAS (slightly positive) and the lowest FGE. This demonstrates the
advantage of adopting a model ensemble rather than the prediction provided by individual models.
Most models underestimate $O_3$ (likely as a result of the overestimated $NO_2$ because the $O_3$
production is not NOx-limited) during the whole period under consideration. For $O_3$, the CHIMERE
model shows slightly better performance (lowest RMSE) than the model ensemble median. The
median BIAS for $O_3$ is relatively constant (slightly negative). For this particular species, the model
ensemble median does not provide the best results regarding the BIAS. In fact, in this case, the
model LOTOS-EUROS gives the best performance for ozone, Interestingly, this particular model
has the largest negative BIAS for $NO_2$. The median BIAS of $O_3$ remains relatively constant during
the period, while the MNBIAS exhibits higher negative values during the winter months, as a result
of the relative low $O_3$ concentrations during winter time.
As stated above, the MarcoPolo-Panda prediction system has the tendency to overestimate surface
$NO_2$, which leads to $O_3$  titration especially during night time. The emission injection height is also
a relevant factor here since it can largely influence the results in the planetary boundary layer.
During night-time, emissions from stacks may be take place above the mixing layer and explain
model-data discrepancies since the models often assume that the injection of primary pollutants
takes place in the first layer above the surface.
Anthropogenic emissions of primary pollutants are changing extremely rapidly in China. The
adopted emissions inventories usually reflect to the situation a few years before the period during
which the model simulations were performed. Since the recent $NO_X$ emissions have decreased
significantly in some urban areas of China in response to measures taken by the local authorities (*F.
Liu et al., 2017*), the anthropogenic emissions used for the current forecasts may be overestimated
in some areas. Some models use reduced $NO_X$ and $SO_X$ anthropogenic emissions (for details see
*Brasseur et al., 2018*), however, daytime concentrations of ozone are generally underestimated in
most models, even when the level of $NO_2$ is in reasonable agreement with the observational values.
The discrepancy could be caused by an underestimation of the emissions of some VOCs, especially
in the center of urban areas where ozone is often VOC-limited.
For PM10 and PM2.5, the model ensemble median shows the best performance compared to all
individual models during the time period under consideration (see Figure 3). For PM10, there is an
overall slight underestimation by all models except by CHIMERE and hence, by the median of the
model ensemble.  For PM2.5, the BIAS is relatively constant (apart in the WRF-Chem-SMS model
which exhibits a lot of variation in the BIAS of PM10 and PM2.5). In this case, the BIAS is slightly
overestimated, but close to zero.



Figure 4 shows the temporal correlation coefficients for $NO_2$, $O_3$, PM10 and PM2.5 for each
individual month, and for the whole time period. It can be seen, that there is a wide spread between
the individual models: the calculated correlations range from 0.2 to 0.7 for $NO_2$, PM10 and PM2.5
and from 0.3 to 0.8 for $O_3$. The model ensemble median and CHIMERE are characterized by high
correlation coefficients in the case of $NO_2$, $O_3$ and PM2.5. For PM10, the model ensemble median
and the LOTOS-EUROS model provide the highest correlation coefficients. In general, the model
ensemble median gives the best performance.
The correlation coefficient of $O_3$ for the ensemble median remains relatively unchanged during the
whole time period, and ranges between 0.6 and 0.8. Considering the whole time period, it is of the
order of 0.75, with CHIMERE providing a slightly higher correlation coefficient for the whole time
period, and also for each individual months. All models exhibit small correlation coefficients in
March 2017. High correlation coefficients are found during the early summer months (June/July).
For PM10 and PM2.5 the correlation coefficients exhibit more variability, starting with very low
correlation for all models and for the ensemble during April and May 2016, high correlation from
June 2016 to March 2017, and again low correlation during April and May 2017. These differences
may be due to missing sources of biomass burning or dust or to individual model tunings. For the
entire time period, the correlation coefficient of the ensemble mean is higher than for each
individual models (~0.58 for PM10 and ~0.78 for PM2.5). The correlation between the model
ensemble and the observations is therefore relatively satisfactory.

## 3.2 Evaluation of the Geographical Distribution
The statistical indicators, described above for all contributing cities, have also been calculated for
the individual cities. The purpose here is to assess regional characteristics and to identify model
issues. Figure 5 shows the statistical indicators (RMSE, BIAS and correlation coefficient) for $O_3$,
$NO_2$ and PM2.5 of the Ensemble Median for each city during the time period under consideration
(April 2016 until June 2017). In the upper most left panel, the BIAS of ozone for each city is
shown. It can be seen, that the ensemble median is underestimating the ozone concentrations in the
north and northeastern regions of China, while no significant bias compared to the observations is
found in cities in the southern part of the country. RMSE in the northern/northeastern cities are
higher (around 40 $\mu g\ m^{-3}$) than in southern and western cities (around 20-30 $\mu g\ m^{-3}$).

The temporal correlation coefficients for ozone calculated for each city over the whole period under
consideration are slightly higher in the northern part of the country and slightly smaller in the
southern regions. This indicates that the day-to-day variability is well simulated, even though the
models are slightly underestimating the ozone pollution in the north. $NO_2$ concentrations (see the
middle panels of Figure 5) are overestimated in some cities and underestimated in other cities.
There is, however, no systematic geographical characterization of the bias. When considering
individual cities, it can be seen that the $NO_2$ concentrations are slightly overestimated in most urban
areas including Beijing, Shanghai, Chengdu, Wuhan and Changsha. The RMSE for $NO_2$ in the
middle panel of Figure 5 is very uniform (around 20 $\mu g\ m^{-3}$) in the whole country. The correlation
coefficients of $NO_2$ (between 0.5 and 0.7) are smaller than those of $O_3$, as $NO_2$ exhibits more
temporal variability than $O_3$. In the case of PM2.5, (see upper most right panel), the concentrations
are well simulated in the northern and southern parts of China, but there are a few city clusters in
the middle of the domain (Chengdu, Chongqing, Wuhan and Changsha) in which the PM2.5
concentrations are overestimated by more than 50$\mu g\ m^{-3}$. These cities also show overestimation
of $NO_2$. The overestimation of PM2.5 may therefore be related to the errors in precursor emissions,
e.g. $NO_X$, $SO_2$. The RMSE of PM2.5 is smaller in the southern part of the domain and along the

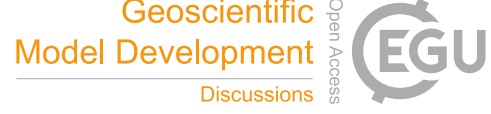



coastline of China, while the model results are less satisfactory in the city clusters located in the
central part of the domain, with very high RMSE of 60-80µg m$^{-3}$ in three cities. The correlation
coefficients for the individual cities are relatively constant around 0.7 with few cities characterized
by lower correlation coefficients (mostly in the central part of the domain).

### 321  3.3 Evaluation of the diurnal variation

We now examine the ability of the models to reproduce the diurnal variations of the chemical
species' concentrations. We first provide a general view based on all observations in China and then
examine the particular situation in the city of Beijing.

### 326  3.3.a Analysis based on all observations in China

The RMSE, BIAS, MNBIAS, and FGE of O3, NO2, PM10 and PM2.5 for the seven models and the
ensemble median for all available observations in China are displayed over the forecasting time (0-
23h) (Figure 6 and 7). Due to storage limitations, only the predictions for the first 24 hours (0-23h)
were saved while the predictions for the 24h-72h period performed by all models were not retained.
Unfortunately, this does not allow the investigation of a day to day degradation of the statistical
indicators (from day1 to day3). Only the diurnal behavior of the statistical indicators can be
assessed, which provides important hints for possible model issues.

It can be seen in the left panels of Figure 6 that the statistical indicators of NO$_2$ for the ensemble
median is relatively stable over the time of the day, with slightly higher RMSE and higher
BIAS/MNBIAS during the night time hours. For the individual models, the variability of the RMSE
is somewhat higher during daytime, while some models exhibit very high RMSE and BIAS during
the night time hours. Most models show a positive BIAS of NO$_2$ during the night, but a few of them
exhibit a negative bias; this results in a relatively small BIAS for the ensemble median, showing
good results with respect to the BIAS throughout the day.

In the case of ozone, the statistical indicators exhibit a variation over the time of the day. The
RMSE is smallest between 7:00 and 9:00 local time, after which it increases until 18:00 in the
evening to become constant at about 30 µg m$^{-3}$ during the night.

An examination of the BIAS and MNBIAS for O$_3$ over the day shows that O$_3$ is underestimated by
nearly all models, apart from WRF-Chem-SMS. This might result from the slight overestimation of
NO$_2$ concentrations by most models. Especially during nighttime when the height of the boundary
layer is low, near surface NO$_2$ concentrations are high, and ozone is underestimated by 50% – 100%
by most models. In the first hours of the day, only SILAMtest, WRF-Chem-SMS and LOTOS-
EUROS exhibit slightly positive O$_3$ BIAS. The same models produce a negative BIAS for NO$_2$
during the first hours of the day.

Figure 7 shows that the BIAS and MNBIAS of both PM10 and PM2.5 stay relatively constant over
the time of the day. PM10 is slightly underestimated by the ensemble median (-5 to -10%), while
PM2.5 is slightly overestimated (10 to 25%). In most cases, the models overestimate the PM2.5
observations, while for PM10 there are stronger differences between the individual models.

For PM10 and PM2.5, the ensemble median exhibits a better performance than the individual
models: the RMSE BIAS, MNBIAS and FGE of the ensemble are on average lower than the





corresponding statistical parameters of the individual models. This demonstrates again the
advantage of using the ensemble median for the prediction of PM10 and PM2.5.

Figure 8 presents the diurnal variation of the concentrations of $O_3$, $NO_2$, $O_3 + NO_2$ and PM2.5 from
the individual models (and the ensemble median) and from the observations at a specific location
(Beijing). The RMSE and the BIAS are also provided during the whole period under consideration.

It can be seen that the ensemble median (black line) underestimates the $O_3$ observations (red line)
throughout the day, especially during the nighttime hours and in the late afternoon. Only WRF-
Chem-SMS reproduces the amplitude of the $O_3$ diurnal cycle, but it also underestimates the $O_3$
concentrations after 18:00 when the height of the boundary layer is rapidly decreasing. All models
and the ensemble median reproduce the diurnal cycle with a maximum in the late afternoon, but this
maximum produced by the model appears about 2 hours earlier than observed. When considering
the RMSE, the models produce the best results during the morning, and with increasing $O_3$
concentrations as the day progresses, the RMSE is also increasing. The negative BIAS is increasing
for all models and for the model ensemble throughout the day.

## 3.3.b Analysis for the specific case of Beijing

In Beijing, the diurnal variation of the $NO_2$ concentrations is overestimated by the individual
models as also reflected by the ensemble median. During the nighttime, for example, the observed
concentrations are about 20-30 $\mu g\ m^{-3}$ lower than the concentrations associated with the ensemble
median. The individual models and the ensemble median show a much stronger diurnal behavior
than the observations. Atmospheric measurements suggest that the concentrations of $NO_2$ are
relatively constant over the time of the day. This might be due to applied temporal profiles of the
anthropogenic emissions or issues in the vertical mixing of the individual models. Also, the models
with their spatial resolution may not capture the details seen in the observations by the ground
network. The RMSE of all models and for the ensemble median is highest in late afternoon and
during the night. The MarcoPolo-Panda prediction system has thus a tendency to overestimate
surface NO2, which leads to an overestimation of the O3 titration especially at night.

To further analyze the chemical coupling between ozone and $NO_2$, we have added at each time step
the mixing ratios of $O_3$ and $NO_2$. The resulting variable, called Ox and expressed here in ppbv, has
the advantage of not being affected by the fast interchange (null cycle) and the resulting partitioning
between ozone and $NO_2$ produced by reactions $NO + O_3$, $NO_2 + h\nu$ and $O + O_2 + M$. If only these
three rapid photochemical reactions are considered, Ox is a conserved quantity. In other words,
even when a more comprehensive chemical scheme is adopted, the diurnal cycle of Ox should be
considerably less pronounced that the diurnal cycle of $NO_2$ and $O_3$.

In fact, in the model forecasts, the sum of $O_3$ and $NO_2$, is nearly constant during the day, but
exhibits nevertheless some diurnal variation, which appears to be weaker than in the observation.
The calculated $O_X$ is slightly too high at night and too low during daytime, suggesting an
overestimation in photochemical activity by the majority of the models. The partitioning of $O_X$ into
$NO_2$ and $O_3$ is not well reproduced despite the simple chemistry that determines this partitioning:
$NO_2$ is generally too high and $O_3$ too low, especially in the afternoon and early night. The simple
partitioning approach does not seem to work properly under high $NO_X$ loading. As a result, the
diurnal cycle of $O_3$ is not well reproduced by the forecasting ensemble and high ozone events are
generally underestimated. This issue is discussed in more detail in the companion paper by
*Brasseur et al., 2018*.

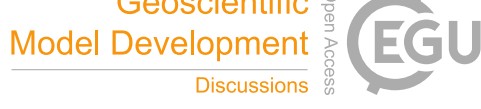



The observed diurnal variation of PM2.5 is not well reproduced by the models and by the ensemble
median. The calculated variability in Beijing is substantially higher than suggested by the
observations (which are characterized by relatively constant concentrations throughout the day).
The models show a maximum in PM2.5 concentrations around 8-9 a.m., and a second maximum
during nighttime hours. This morning maximum is not present in the observations.  The model
ensemble is overestimating the observations in the morning and underestimating them in the early
afternoon, resulting in a diurnal variability of the BIAS, shown in the lowest panel. Again, this
might be related to the adopted diurnal profiles of the anthropogenic emission sources or might be
due to errors in the formulation of vertical mixing in the PBL.

## 4. The impact of missing model data on the ensemble performance
To assess the impact on the ensemble forecast of occasionally missing results from one or several
models, we compare the following ensembles during a given test period (1-30 May 2017),
separately for $O_3$, $NO_2$ and PM2.5: This approach has already been adopted by *Marécal et al., 2015,*
to evaluate European air quality predictions. We consider the following cases*:*
- "MEDIAN 7", the median provided by the operational ensemble method, which includes all seven
models;
- "MEDIAN 5", the median built on five individual models, excluding the "best" and the "worst"
models;
- "MEDIAN 3", the median built on three individual models, excluding the two "best" and the
"two" worst models;
- "BEST", the model with the highest performance;
- "WORST", the model with the lowest performance.
Since the relative performance of individual models varies in time and space, the criterion to order
the seven individual models from "worst to best" is provided by the value of their respective RMSE
over the test period. For ozone, the criterion is measured by the RMSE over the 30 days between
12:00 and 18:00 LST (ozone peak time) (this criterion is based on the fact that the "best" model
refers to the best forecast of daytime ozone levels).  RMSE is seen as the most objective criterion
since MB and MNMB can include compensating effects.
Figure 9 shows the statistical indicators for May 2017 as a function of the forecasting time (0-23h)
of the ensemble median based on all 7 models (MEDIAN7, shown in red), 5 models (MEDIAN5,
shown in blue), and 3 models (MEDIAN3, shown in black). The results are also shown for the
"best" and the "worst" model (BEST (magenta) and WORST (light blue)). For all three species, the
ensemble median based on 7 models is of highest quality (based on the statistical indicators used in
this analysis), and generally surpasses the results provided by the "best" model. When only 5
models (excluding the best and the worst) are available to calculate the ensemble, all statistical
indicators show only very small differences with the more inclusive MEDIAN7 case based on seven
models. Reducing the ensemble calculation further to three models (MEDIAN3), the statistical
scores degrade slightly compared to the MEDIAN7 and MEDIAN5 for all three species, but remain
higher or at least similar to the score of the "best" model (BEST).
It is interesting to note that the "best" model (BEST) is not the same model for the different months
that are investigated, nor the same model for all species. For example, in August 2016, the "best"



model for $O_3$ and PM2.5 is IFS, while LOTOS-EUROS shows the best performance for $NO_2$. In
May 2017, the best model for PM2.5 is LOTOS-EUROS and the worst model is IFS, but the results
remain the same: the ensemble product performs better than (or at a similar level as) the best model.
Since the "BEST" model can change depending on time period and species, the ensemble product is
particularly valuable for the sustained quality of the forecasting system. This study shows therefore
that using the ensemble product (median) of models, even if occasionally based on fewer models, is
more useful than using a single model, even if the performance of this individual model is high. The
ensemble product is still robust compared to the observations if the output of some contributing
models is occasionally missing. It also shows that an ensemble product remains valuable even if
only few models are available for the production of the forecast.

## 5. Performance of the Forecasting System for Alert Warnings

The prediction system has been designed to support the development of policies and the calculation
of air quality indexes. One of the applications of the system is to provide alerts to the general public
when acute air pollution episodes are expected. Thus, the performance of the forecast system has
been tested regarding the likelihood to predict air pollution events. We will refer to this type of
forecast as binary prediction of events (*Brasseur and Jacob, 2017*).
A model prediction of a specific event such as an air pollution episode at a given location (e.g.
concentration of pollutants exceeding a regulatory threshold) is evaluated by considering a binary
variable and by distinguishing between four possible situations: (1) the event is predicted and
observed, (2) the event is not predicted and not observed, (3) the event is predicted but not
observed, (4) the event is not predicted but is observed. Cases (1) and (2) are regarded as successful
predictions (hits), while (3) and (4) are considered to be failures (misses). The skill of the model for
binary prediction (event or no event) is measured by the fractions of observed events that are
correctly predicted (probability of detection (POD)). The fraction of predicted events, that did not
occur is measured by the false alarm rate (FAR)).
We have calculated the POD and the FAR for the ensemble median for the cities of Beijing,
Shanghai and Guangzhou between April 2016 and June 2017, specifically for ozone (based on the 8
hour and the daily maximum value), $NO_2$ and PM2.5. The air quality indexes are calculated for 1)
1-hour ozone, 2) 8-hour ozone concentrations 3) 24-hour mean $NO_2$ concentrations, 4) 1-hour $NO_2$
concentrations and 5) 24-hour PM2.5 concentrations. The definitions breakpoints for the individual
air quality indexes (AQI) are shown in Table 1 and Table 2; they are based on current definitions of
AQI from the Chinese government.
**Table 1:** Chinese AQI categories

| Index values | AQI levels | AQI categories |
|---|---|---|
| 0-50 | 1 | Good |
| 51-100 | 2 | Moderate |
| 101-150 | 3 | Lightly polluted |
| 151-200 | 4 | Moderately polluted |
| 201-300 | 5 | Heavily polluted |
| >300 | 6 | Severely polluted |



**Table 2:** Individual AQI for 1-hour and 8-hour Ozone, 24-hour and 1-hour NO$_2$ and 24-hour PM2.5

| IAQI | 1-hour O$_3$ [µg m$^{-3}$] | 8-hour O$_3$ [µg m$^{-3}$] | 24-hour NO$_2$ [µg m$^{-3}$] | 1-hour NO$_2$ [µg m$^{-3}$] | 24-hour PM2.5 [µg m$^{-3}$] |
|---|---|---|---|---|---|
| 0 | 0 | 0 | 0 | 0 | 0 |
| 50 | 160 | 100 | 40 | 100 | 35 |
| 100 | 200 | 160 | 80 | 200 | 75 |
| 150 | 300 | 215 | 180 | 700 | 115 |
| 200 | 400 | 265 | 280 | 1200 | 150 |
| 300 | 800 | 800 | 565 | 2340 | 250 |
| 400 | 1000 | Use hourly | 750 | 3090 | 350 |
| 500 | 1200 | Use hourly | 940 | 3840 | 500 |

In order to highlight the presence of thresholds violated during the time period under consideration,
Figure 10-12 show the time series for the period April 2016 – July 2017 of the 1) daily maximum
ozone concentrations, 2) 8-hour moving average of ozone, 3) the 24-hour mean NO$_2$ concentrations,
4) the daily maximum NO$_2$ concentrations and 5) the 24-hour mean PM2.5 concentrations for
Beijing (Figure 10), Shanghai (Figure 11) and Guangzhou (Figure 12) derived from the model and
from the observations at each location. Pink lines indicate the thresholds for the air quality indexes
for moderate (line), lightly polluted (dashed line) and moderately polluted (dotted line) conditions
for each pollutant.
In Beijing and Shanghai, the daily maximum ozone concentrations exceeded the thresholds of 160
(moderate) and 200 (lightly polluted) within the considered time period only during the months of
April to September 2016. During the months of October 2016 to March 2017, the ozone
concentrations remained below the threshold of 160, highlighting fair air quality conditions with
regard to ozone in wintertime. In Beijing, the ensemble median has a probability of detection of air
pollution events for moderate 1-hour ozone AQI of 0.44 (55 out of 126 events of 1-hour ozone
breaking the threshold of 160 µg m$^{-3}$ have been detected). The False Alarm Rate (FAR) is 0.05 (the
model ensemble predicted 58 events where ozone exceeds the threshold of 160 µg m$^{-3}$, where 3 out
of these 58 events were false alarm (observations below the threshold). Lightly polluted events (1-
hour ozone exceeding 200 µg m$^{-3}$) were correctly predicted only 14 times, while the observations
exceeded the threshold 79 times. The FAR for lightly polluted ozone events is 0.12 (2 out of 16).
For moderately polluted ozone events (1-hour ozone exceeding 300 µg m$^{-3}$), the POD is 0, the
model ensemble was not able to predict the 4 observed events (FAR is not applicable, (0 out of 0)).
Looking at the 8-hour ozone predictions for Beijing, the model ensemble is very similar, with a
POD of 0.45 (864 out of the 1921 observed events have been predicted correctly) and a FAR of
0.06 (56 counts are false alarm out of 920 events). For lightly polluted ozone conditions, the POD is
0.18 (118 out of 657 observed events) with a FAR = 0.06 (7 out of 125 are false alarm). For
moderately polluted conditions, the model ensemble predicted 7 out of 150 observed events
correctly with a FAR of 0.22 (2 out of 9 alarms are false).
For Shanghai, the PODs for ozone predictions are lower than in Beijing: for moderate air quality
conditions, the POD is 0.16 (15 out of 92 observed events are predicted correctly) with a FAR of 0



(no false alarm) for 1-hour ozone predictions, and POD = 0.21 (488 out of 2346 observed events)
with a FAR of 0.01 (7 false alarms relative to 495 counts) for 8-hour ozone predictions. For lightly
polluted conditions, the POD is decreasing: POD = 0.08 (3 correct predictions out of 38 observed
events) with FAR of 0 (no false alarm, 3 correct predictions) for 1-hour ozone, and POD = 0.07 (27
out of 398 observed) with a FAR of 0.10 (3 false alarms out of 30) for 8-hour ozone. For
moderately polluted conditions (1-hour ozone exceeding 300 µg m$^{-3}$ or 8-hour ozone exceeding 215
µg m$^{-3}$), the POD for 1-hour ozone is not applicable (no predicted, no observed events), and for 8-
hour ozone POD = 0 (0 predicted out of the 29 observed), FAR = 1 (2 false alarms out of 2
predicted, but not observed).
In Guangzhou, there is no clear difference between ozone conditions in summer or wintertime
during the considered time period. Ozone observations regularly exceed the threshold of 160
(moderate) and 200 µg m$^{-3}$ (lightly polluted) during the whole time period, and 5 times 1-hour
ozone is exceeding the threshold of 300 µg m$^{-3}$.
The POD of 1-hour ozone in Guangzhou is 0.16 (15 correct predictions out of 94 observed) with
FAR = 0.21 (4 false alarms out of 19 predicted) for moderate conditions, and POD = 0.03 (1
predicted out of 36 observed) with FAR = 0 (0 out of 1 predicted) for lightly polluted conditions,
and POD = 0 (0 predicted out of 5 observed events) for moderately polluted ozone conditions. For
8-hour ozone, the POD is 0.31 (315 correct predicted out of 1032 observed) with FAR = 0.28 (122
false alarms of 437 predicted events) for moderate conditions, POD = 0.06 (12 out of 217 observed)
with FAR = 0 (no false alarm out of 12 predicted events) for lightly polluted ozone conditions, and
POD = 0 (0 out of 47 observed events) for moderately polluted ozone conditions.
In general, the ability of the model ensemble to predict correctly ozone air pollution events is best
for light ozone pollution, while it fails to predict correctly the ozone pollution events for moderately
polluted situations. This is mostly a result of the model ensemble being too low compared to the
observations. The predictions can be improved by applying a bias correction to the ozone
predictions. This is investigated in the following Section 5.1.
The NO$_2$ predictions of the ensemble median are in general too high compared to the observation,
especially in Beijing and Shanghai. Especially, in summertime (June/July/August/September), the
model predictions are sometimes twice as high as the observations, which might be a result of
uncertainties in the emissions. In all three cities under consideration, the NO$_2$ concentrations are
only exceeding the thresholds of 40 µg m$^{-3}$ for 24-hour NO$_2$ (100 for 1-hour NO$_2$) and 80 µg m$^{-3}$ for
24-hour NO$_2$ (200 µg m$^{-3}$ for 1-hour NO$_2$) during the considered period (moderate and lightly
polluted conditions for NO$_2$). During wintertime (November/December/January), the observations
are slightly higher than in summer and the ensemble system is in better agreement with the
observations.
In Beijing, the POD for 24-hour NO$_2$ is 1 (214 of 214 observed events are predicted) for moderate
conditions with a FAR of 0.46 (180 false alarms relative to 394 predicted events). This indicates
that NO$_2$ is generally overestimated by the model ensemble. For lightly polluted events, the POD is
0.79 (27 predicted out of 34 observed events) with FAR = 0.70 (63 false alarms out of 90
predicted).  For the 1-hour NO$_2$, the POD for moderate conditions is 0.61 (36 out of 59 observed
events) with FAR = 0.80 (141 false alarms out of 177 predicted). For lightly polluted conditions, no
events have been observed nor predicted for 1-hour NO$_2$ in Beijing during the considered period. In
Beijing, the threshold for moderately polluted NO$_2$ conditions has not been exceeded neither by 1-
hour NO$_2$ nor by 24h- NO$_2$ during the considered period.

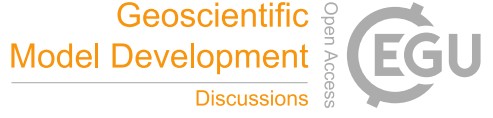

In Shanghai, the numbers are very similar to those in Beijing: POD for 24-hour $NO_2$ is 1 (208 of
208 observed events are predicted) for moderate conditions with a FAR of 0.42 (152 false alarms of
360 predicted events). There is also a general overestimation by the model ensemble compared to
the observations. For lightly polluted conditions, the POD for 24-hour $NO_2$ is 0.67 (10 out of 15
observed) and a FAR of 0.86 (60 false alarms of 70 predicted), which is a clear result of the
overestimated $NO_2$. For the 1-hour $NO_2$, the POD is 0.91 (48 predicted out of 53 observed) with a
FAR of 0.70 (111 false alarms out of 159 predicted) for moderate conditions. The thresholds for
lightly polluted and moderately polluted conditions for 1-hour $NO_2$ have not been exceeded in
Shanghai during the considered period, but there was 1 false alarm (1 out of 1) for lightly polluted
conditions.
In Guangzhou, the model ensemble and the observations for $NO_2$ are in better agreement. There is
slight overestimation of the $NO_2$ concentrations from May to September 2016, and in May 2017,
but in general, there is a good agreement between the model time series and the observations. The
POD for 24h-$NO_2$ exceeding the threshold for moderate conditions is 0.94 (208 predicted out of 222
observed) with a FAR of 0.35 (110 false alarms of 318 predicted events), for lightly polluted
conditions POD is 0.56 (15 predicted out of 27 observed) with 32 false alarms out of 47 predicted
events (FAR = 0.69). Stronger polluted events have not been observed nor predicted for $NO_2$ in
Guangzhou. For the 1-hour $NO_2$, 58 events have been predicted out of 76 observed for moderate
conditions (POD = 0.76, FAR = 0.63 (97 false alarms out of 155 predicted). For lightly polluted
conditions, there was 1 false alarm (1 out of 1), with neither observed nor correctly predicted
events.
The thresholds for moderately polluted conditions for 24-hour $NO_2$ and 1-hour $NO_2$ have not been
exceeded in Guangzhou during the considered period, no events have been predicted nor observed.
The predictions of PM2.5 (24-hour PM2.5) of the model ensemble are in very good agreement with
the observations in all three cities during the considered period.
In Beijing, the POD for the prediction of moderate condition for 24-h PM2.5 is 0.95 (268 correctly
predicted events out of 283 observed) with a FAR of 0.19 (61 false alarms out of 329 predicted
events). For lightly polluted conditions, the POD is 0.76 (111 correct predicted events of 146
observed events) with a FAR of 0.28 (43 false alarms for 154 predicted events). Moderately
polluted PM2.5 events have been correctly predicted 33 times out of 64 observed events (POD =
0.52) with a FAR of 0.35 (18 false alarms out of 51 predicted events).
In Shanghai, 191 moderate condition-events for PM2.5 have been correctly predicted out of 220
observed events (POD = 0.87, FAR = 0.19), with 46 false alarms out of the 237 predicted events.
For lightly polluted events, the POD is 0.84 (32 out of 38 observed events) with a FAR of o.47 (28
false alarms of 60 predicted events). For moderately polluted conditions of PM2.5, the POD is 0.50
(3 correctly predicted events out of 6 observed) with a relatively high FAR (0.67, 6 false alarms out
of 9 predicted).
In Guangzhou, the POD for moderate conditions of PM2.5 is 0.85 (149 correctly predicted out of
175 observed) with 65 false alarms out of 214 predicted events (FAR = 0.30). Lightly polluted
events have been observed only 7 times, the ensemble median predicted 4 of them correctly (POD =
0.57), but with a very high false alarm rate (16 false alarms out of 20 predicted events, FAR =
0.80), this indicates a slight overestimation of the PM2.5 concentrations of the models compared to
the observations. In Guangzhou, no moderately polluted events of PM2.5 have been observed nor
predicted during the considered period.



Only in Beijing, and only with regard to 24-hour PM2.5, heavily polluted conditions have been
observed and predicted during the considered period in the winter months 2016/2017: The POD is
0.5 (18 correct predicted out of 36 observed events) with a FAR of 0.28 (7 false alarms out of 25).
These investigations show, that the model ensemble is well suited to be used in air quality
predictions of PM2.5. For ozone, due to biases of the model ensemble compared to observations,
the model ensemble is not able to predict ozone pollution in an appropriate way. Although the FAR
is very low for ozone predictions, the POD of model ensemble is not very high. In the following
Section, we apply bias correction to improve the predictions for ozone pollution events.

## 5.1 Bias Correction for Ozone Predictions

Bias corrections can be applied to improve the predictions of an individual model or a model
ensemble. In our case, we have calculated the summertime bias of the time series of the hourly
ozone concentrations from the model ensemble with respect to the hourly observations, and
subtracted the bias from the hourly time series. For predictions of ozone air pollution, the
summertime is an appropriate season to consider since the ozone thresholds are exceeded only
during this season. As the bias between the observations and the model might not be the same for
each month, and our goal is to obtain the best improvement in the ozone predictions for
summertime, we have subtracted the mean summertime bias (mean of the bias of
June/July/August/September 2016) from the original time series. The daily maximum ozone values
and the 8-hour moving average for the corrected time series have then been calculated. The
resulting, POD and FAR for 1-hour ozone and 8-hour ozone under different air quality conditions
are shown in Table 3. This table shows that, for bias-corrected predictions, the POD in all three
cities is larger than for the non-corrected time series, especially in the case of moderate and lightly
polluted conditions of ozone. Thus, the predictions of air pollution events are significantly
improved when the bias correction is applied in the case of ozone. Only for the predictions of
moderately polluted conditions of ozone, the POD is not changing. The FAR is also slightly
decreasing for all cities, but the improvement is small.
In Beijing, the POD air pollution events represented by a moderate AQI for 1-hour ozone increased
from 0.44 for Beijing (55 out of 126 observed events) before bias correction to 0.69 (87 out of 126
events) after bias correction. The False Alarm Rate (FAR) also increased from 0.05 (3 false alarms
out of these 58 events) to 0.10 (10 false alarms out of 97 predicted events). Lightly polluted events
(1-hour ozone exceeding 200 µg m$^{-3}$) have been predicted correctly 31 times (14 times without the
corrections), while the observations exceeded the threshold 79 times. The FAR for lightly polluted
ozone events also slightly increased from 0.125 (2 out of 16) to 0.2 (8 false alarms out of 40).
For moderately polluted ozone events (1-hour ozone exceeding 300 µg m$^{-3}$), the POD for the bias-
corrected prediction is still 0. The model ensemble was not able to predict the 4 observed events
(FAR is not applicable, (0 out of 0)).
Looking at the 8-hour ozone predictions for Beijing, the POD of 0.45 (864 out of the 1921 observed
events have been predicted correctly) increased to 0.76 (1452 out of 1921) after bias corrections,
and the FAR from 0.06 (56 counts are false alarm out of 920) to 0.23 (424 false alarms out of 1876
predictions) for moderate ozone pollution. For lightly polluted ozone conditions, the POD increased
to 0.44 (291 out of 657) and FAR = 0.22 (81 false alarms of 372 predicted) for the bias corrected
predictions compared to POD = 0.18 (118 out of 657 observed events) with a FAR = 0.06 (7 out of
125 are false alarm). For moderately polluted conditions, the model ensemble with bias corrected



predicted 27 (instead of only 7) out of 150 observed events correctly with a FAR of 0.28 (13 false
alarms of 47 predictions) compared to FAR of 0.22 (2 out of 9 are false alarm).
For Shanghai, for moderate air quality conditions of ozone, the POD increased from 0.16 to 0.51
(47 (15 for non-corrected) out of 92 observed events are predicted correctly); the FAR increased
from 0 (no false alarm) to 0.10 (5 false alarms out of 52) for 1-hour Ozone predictions. For 8-hour
ozone predictions, the POD increased from 0.21 to 0.66 (1554 (non-corrected: 488) out of 2346
observed events), the FAR  increased from 0.01 (7 false alarms of 495 predicted events) to 0.32
(726 false alarms of 2280 counts) for 8-hour ozone predictions. For lightly polluted ozone
conditions, the POD increased from 0.08 (3 correct predictions out of 38 observed) with FAR of 0
(no false alarm, 3 correct predictions) to POD = 0.34 (13 out of 38) with FAR = 0.07 (1 false alarm
of 14 predicted events) for 1-hour ozone, and for 8-hour ozone, the POD increased from 0.07 to
0.27 (109 (non-corrected: 27) out of 398 observed) and the FAR increased from 0.10 (3 false alarms
out of 30) to 0.13 (16 false alarms in 125 predicted events). For moderately polluted ozone
conditions, the POD for 1-hour ozone is not applicable for both non-corrected and bias-corrected
predictions (no predicted, no observed events), but for the bias-corrected prediction, one false alarm
is observed (FAR = 1, 1 false alarm in 1 predicted event), and for 8-hour ozone POD increased
from 0 to 0.10 (3 (non-corrected: 0) predicted out of the 29 observed), the FAR decreased from 1 (2
false alarms out of 2 predicted, but not observed) to 0.8 (12 false alarms of 15 predicted events).
In Guangzhou, the predictions are not as accurate as in Beijing and Shanghai, and the bias
corrections result only in slight improvements of the ozone forecasts for Guangzhou. The POD of 1-
hour ozone in Guangzhou increased from 0.16 to 0.32 (30 (non-corrected: 15) correct predictions
out of 94 observed) and the FAR slightly increased from 0.21 (4 false alarms out of 19 predicted) to
0.33 (15 false alarms out of 45 predicted events) for moderate conditions. For lightly polluted ozone
conditions, the POD increased from 0.03 to 0.14 (5 (non corrected: 1) predicted out of 36 observed)
and the FAR increased from 0 (0 out of 1 predicted) to 0.29 (2 false alarms of 7 predicted events).
For moderately polluted ozone predictions, the POD and FAR did not change with bias corrections
(POD = 0 (0 predicted out of 5 observed events), FAR not applicable).
For 8-hour ozone of moderate conditions, the POD increased from 0.31 to 0.49 (508 (non-corrected:
315) correct predicted out of 1032 observed) and the FAR increased from 0.28 (122 false alarms of
437 predicted events) to 0.37 (296 false alarms for 804 predictions). For lightly polluted ozone
conditions the POD increased from 0.06 to 0.13  (29 (non-corrected: 12) out of 217 observed) and
the FAR increased from 0 (no false alarm out of 12 predicted events) to 0.19 (7 false alarms for 36
predicted events). For moderately polluted ozone conditions, the POD and FAR did not change with
bias corrections (POD= 0 (0 out of 47 observed events), FAR not applicable).
Figure 13 a–c shows the time series of the model ensemble, the bias corrected time series of the
model ensemble and the observations. For the daily maximum ozone, the bias correction results in a
better agreement with the observations, which also results in better event predictions. For 8-hour
ozone, there is better agreement during summertime, while during the wintertime, the bias-corrected
ozone time series are too high compared to the observations (both correcting for the bias derived
from the total time series, or only from the summertime time series). This shows (as we have seen
in Section 3.1), that the bias is not the same during the whole year, and also that the diurnal cycle of
ozone is not well captured by the model ensemble. While the bias corrected daily maximum ozone
is in better agreement with the observations, the 8-hour bias corrected moving average is too high
during winter time (with very low ozone concentrations). As the ozone is too low in winter to
exceed the lowest threshold (moderate conditions) for air quality index calculations, this is not
affecting the quality of the event prediction. A more sophisticated bias-correction (bias correction



with diurnal and annual variation included) could be applied to further improve the predictions,
provided that a longer time series (more than one year of data) is available. The statistical bias
correction can then be used for the improvement of future predictions.

## 6. Conclusions and Future Developments
In this paper, we evaluate the forecasting system developed and implemented as part of the EU
Panda and MarcoPolo projects after a little more than one year of operation. The forecasting system
is based on an ensemble of seven state-of-the-art chemistry-transport models (CHIMERE, EMEP,
IFS, LOTOS-EUROS, WRF-Chem-MPIM, WRF-Chem-SMS, SILAMtest). Each model is
executed on a computer platform hosted by individual institutes in China and Europe. Input for
meteorological forcing, emissions and boundary conditions have been carefully chosen and adopted
for the specific situation of China, but vary from model to model. The forecasting system provides
every day hourly forecasts for 3 days ahead for four major chemical pollutants ($O_3$, $NO_2$, PM10 and
PM2.5) together with hourly observational data provided by the Chinese observational network
(www.pm25.in).
The models, whose predictions are strongly influenced by the adopted weather forecast, reproduce
in general the regional features and capture many air pollution events. In most cases, the model
ensemble reproduces satisfactorily the day-to-day variability of the concentrations of the primary
and secondary air pollutants and in particular, predicts the occurrence of pollution events a few days
before they occur. Overall, and in spite of some discrepancies, the air quality forecasting system is
well suited for the prediction of air pollution events and has the ability to be used for alert warning
(binary prediction) of the general public, specifically if bias corrections are applied to improve the
ozone forecasts.
In most cases, the ensemble approach provides more accurate forecasts and reduces the
uncertainties in comparison with the individual models results. The calculation of the median of all
models is also relatively insensitive to model outliers, and is computationally efficient. Using the
ensemble median based on all models provides the best performance for all species, as the relative
performance of any individual model may vary in time, space and species. We showed, that the
ensemble product, even if occasionally based on fewer models, is more useful than a single model
of good quality, and that the ensemble product is still robust compared to the observations if data
from some contributing models are occasionally missing.
Despite the fact that the prediction system is in its development phase and that the resources
available to improve the system are limited, the MarcoPolo and Panda forecasting system can be
viewed as already quite successful. The inter-comparison presented in the companion paper by
*Brasseur et al., 2018* and the present evaluation were performed to diagnose differences between
models, identify problems and contribute to individual model improvements. Specifically, the
underestimation of ozone under high $NO_X$ conditions and the resulting errors in the diurnal cycle of
ozone need to be addressed in an effort to improve the model forecasts in China. Although major
efforts are ongoing to improve emission inventories for China, the remaining uncertainties,
especially in regard to local emissions, may partly explain the differences between models and
observations. This is subject of further investigation. Furthermore, data assimilation of satellite and
in situ observations should significantly improve the performance of the forecasting system. Finally,
a more advanced approach to extract observations provided by the Chinese network is expected to
improve the model-data comparison.



## Data Availability

The models described here are used operationally by the participating research and service organizations involved in the present study. The data produced by the multi-model forecasting system are available from the Royal Dutch Meteorological Institute (KNMI).

## Acknowledgements

The model inter-comparison presented in the present study has been conducted during a workshop organized in May 2017 by the Shanghai Meteorological Service (SMS) in China. The authors thank Dr. Jianming Xu for hosting this meeting and providing support to the participants. The ensemble of models described here has been produced under the Panda and MarcoPolo projects supported by the European Commission within the Framework Program 7 (FP7) under grant agreements n°606719 and n°606953. The National Center for Atmospheric Research (NCAR) is sponsored by the US National Science Foundation.




**Table 3**: POD and FAR for Beijing, Shanghai and Guangzhou

| | Probability of Detection (POD) | | | False Alarm Rate (FAR) | | |
|---|---|---|---|---|---|---|
| **Beijing** | AQI 2 (moderate) | AQI 3 (lightly poll.) | AQI 4 (moderately poll.) | AQI 2 (moderate) | AQI 3 (lightly poll.) | AQI 4 (moderately poll.) |
| 1-hour $O_3$ [µg m$^{-3}$] | 0.44 (55/126) | 0.18 (14/79) | 0 (0/4) | 0.05 (3/58) | 0.12 (2/16) | NaN (0/0) |
| Bias corrected 1-hour $O_3$ [µg m$^{-3}$] | 0.69 (87/126) | 0.41 (32/79) | 0 (0/4) | 0.10 (10/97) | 0.20 (8/40) | NaN (0/0) |
| 8-hour $O_3$ [µg m$^{-3}$] | 0.45 (864/1921) | 0.18 (118/657) | 0.05 (7/150) | 0.06 (56/920) | 0.06 (7/125) | 0.22 (2/9) |
| Bias corrected 8-hour $O_3$ [µg m$^{-3}$] | 0.76 (1452/1921) | 0.44 (291/657) | 0.23 (34/150) | 0.23 (424/1876) | 0.21 (81/372) | 0.28 (13/47) |
| 24-hour $NO_2$ [µg m$^{-3}$] | 1 (214/214) | 0.79 (27/34) | NaN (0/0) | 0.46 180/394) | 0.70 (63/90) | NaN (0/0) |
| 1-hour $NO_2$ [µg m$^{-3}$] | 0.61 (36/59) | NaN (0/0) | NaN (0/0) | 0.80 (141/177) | NaN (0/0) | NaN (0/0) |
| 24-hour PM2.5 [µg m$^{-3}$] | 0.95 (268/283) | 0.76 (111/146) | 0.52 (33/64) | 0.19 (61/329) | 0.28 (43/154) | 0.35 (18/51) |
| **Shanghai** | | | | | | |
| 1-hour $O_3$ [µg m$^{-3}$] | 0.16 (15/92) | 0.08 (3/38) | NaN (0/0) | 0 (0/15) | 0 (0/3) | NaN (0/0) |
| Bias corrected 1-hour $O_3$ [µg m$^{-3}$] | 0.51 (47/92) | 0.34 (13/38) | NaN (0/0) | 0.10 (5/52) | 0.07 (1/14) | 1 (1/1) |
| 8-hour $O_3$ [µg m$^{-3}$] | 0.21 (488/2346) | 0.07 (27(398) | 0 (0/29) | 0.01 (7/495) | 0.10 (3/30) | 1 (2/2) |
| Bias corrected 8-hour $O_3$ [µg m$^{-3}$] | 0.66 (1554/2346) | 0.27 (109/398) | 0.10 (3/29) | 0.32 (726/2280) | 0.13 (16/125) | 0.80 (12/15) |
| 24-hour $NO_2$ [µg m$^{-3}$] | 1 (208/208) | 0.67 (10/15) | NaN (0/0) | 0.42 (152/360) | 0.86 (60/70) | NaN (0/0) |
| 1-hour $NO_2$ [µg m$^{-3}$] | 0.91 (48/53) | NaN (0/0) | NaN (0/0) | 0.70 (111/159) | 1 (1/1) | NaN (0/0) |
| 24-hour PM2.5 [µg m$^{-3}$] | 0.87 (191/220) | 0.84 (32/38) | 0.50 (3/6) | 0.19 (46/237) | 0.47 (28/60) | 0.67 (6/9) |
| **Guangzhou** | | | | | | |
| 1-hour $O_3$ [µg m$^{-3}$] | 0.16 (15/94) | 0.03 (1/36) | 0 (0/5) | 0.21 (4/19) | 0 (0/1) | NaN (0/0) |
| Bias corrected 1-hour $O_3$ [µg m$^{-3}$] | 0.32 (30/94) | 0.14 (5/36) | 0 (0/5) | 0.33 (15/45) | 0.29 (2/7) | NaN (0/0) |
| 8-hour $O_3$ [µg m$^{-3}$] | 0.31 (315/1032) | 0.06 (12/217) | 0 (0/47) | 0.28 (122/437) | 0 (0/12) | NaN (0/0) |
| Bias corrected 8-hour $O_3$ [µg m$^{-3}$] | 0.49 (508/1032) | 0.13 (29/217) | 0 (0/47) | 0.37 (296/804) | 0.19 (7/36) | NaN (0/0) |
| 24-hour $NO_2$ [µg m$^{-3}$] | 0.94 (208/222) | 0.56 (15/27) | NaN (0/0) | 0.35 (110/318) | 0.68 (32/47) | NaN (0/0) |
| 1-hour $NO_2$ [µg m$^{-3}$] | 0.76 (58/76) | NaN (0/0) | NaN (0/0) | 0.63 (97/155) | 1 (1/1) | NaN (0/0) |
| 24-hour PM2.5 [µg m$^{-3}$] | 0.85 (149/175) | 0.57 (4/7) | NaN (0/0) | 0.30 (65/214) | 0.80 (16/20) | NaN (0/0) |



**Table 4**: POD and FAR for PM2.5 for Beijing under heavily polluted conditions.

| Beijing AQI heavily polluted | POD | FAR |
|---|---|---|
| 24-hour PM2.5 [μg m$^{-3}$] | 0.50 (18/36) | 0.28 (7/25) |





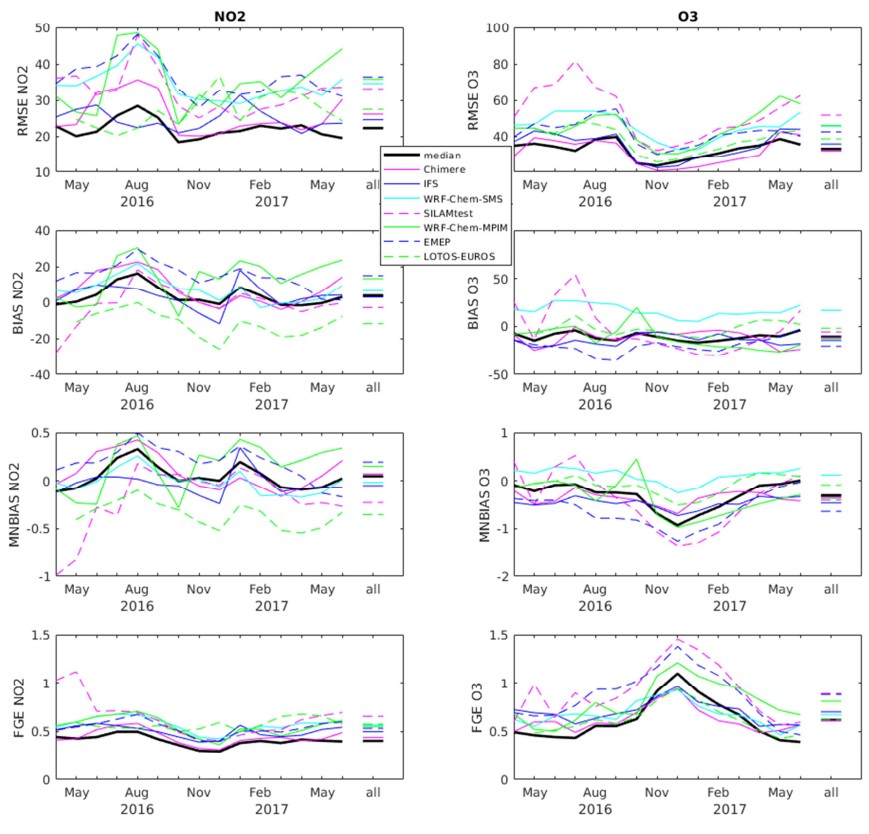

*Figure 2: RMSE, BIAS, MNBIAS and FGE of NO$_2$ and O$_3$ for each month and for the entire time*
*period (April 2016 – June 2017, lines on the right side of each panel).*





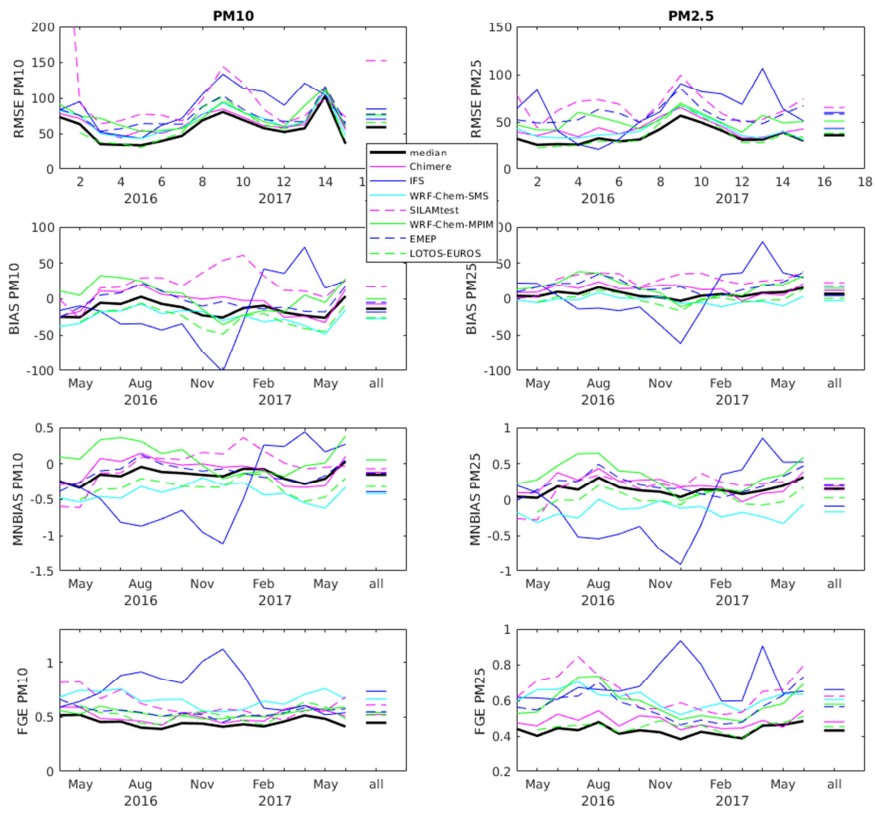

*Figure 3: RMSE, BIAS, MNBIAS and FGE of PM10 and PM2.5 for each month and for the entire*
*time period (April 2016 – June 2017, lines on the right side of each panel).*

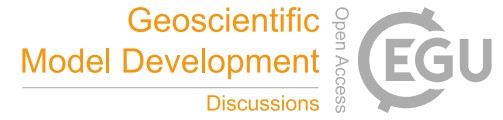



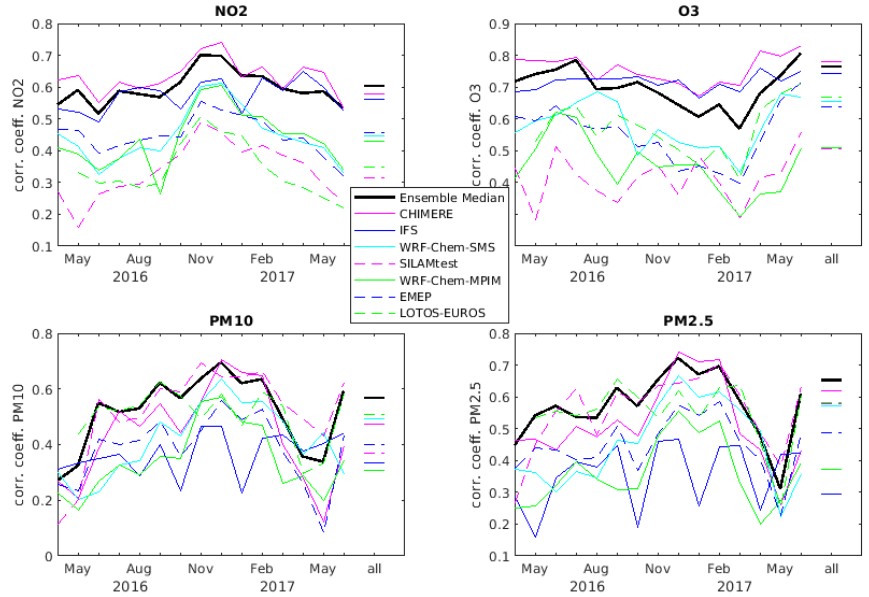

*Figure 4: Correlation coefficients based on hourly concentrations of NO₂, O₃, PM10 and PM2.5 for*
*each month and for the entire time period between April 2016 and June 2017 (lines on the right*
*side of each panel).*



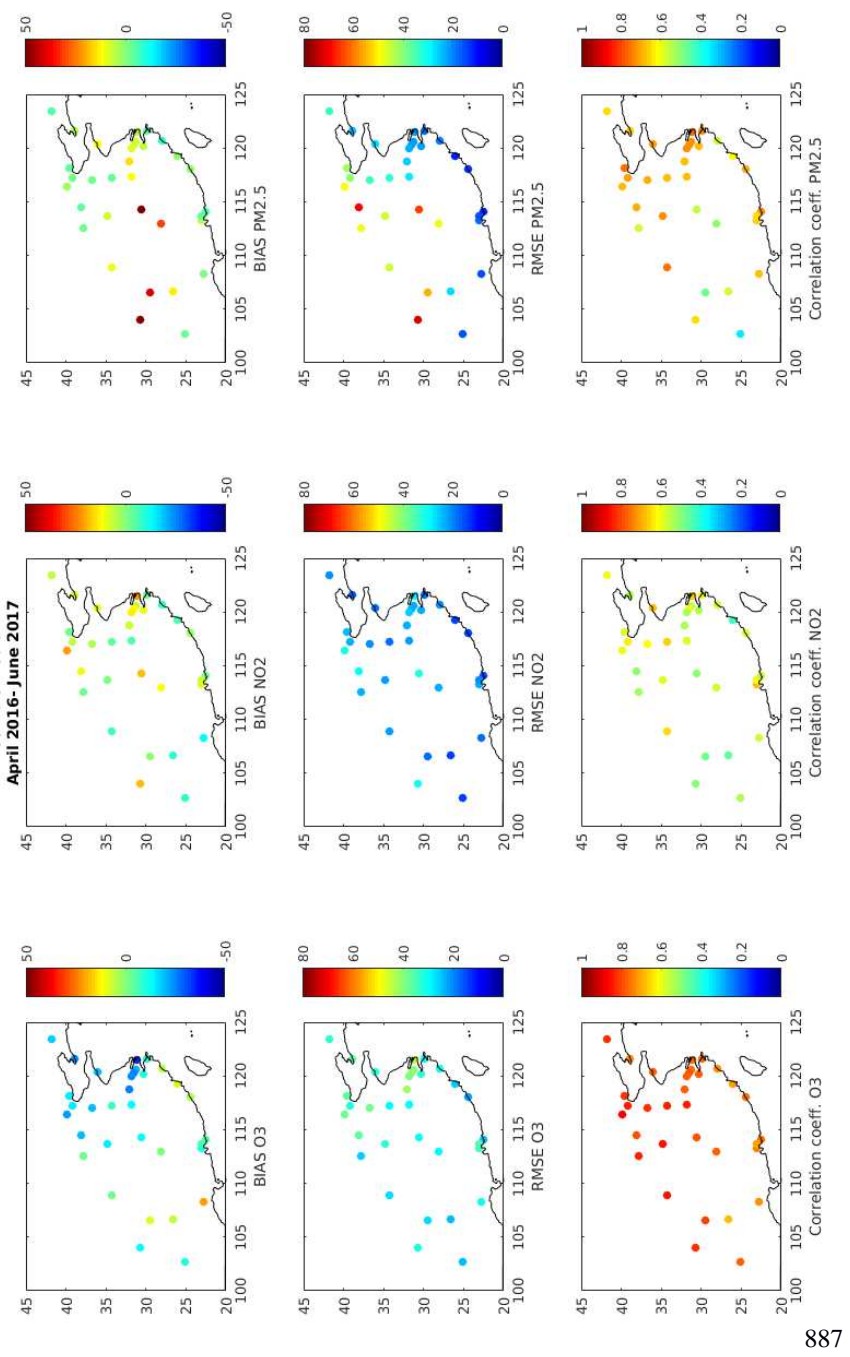


*Figure 5: Map of the BIAS, RMSE and temporal correlation coefficient of O₃, NO₂ and PM2.5 for the whole time period (April 2016 until June 2017) for each city.*





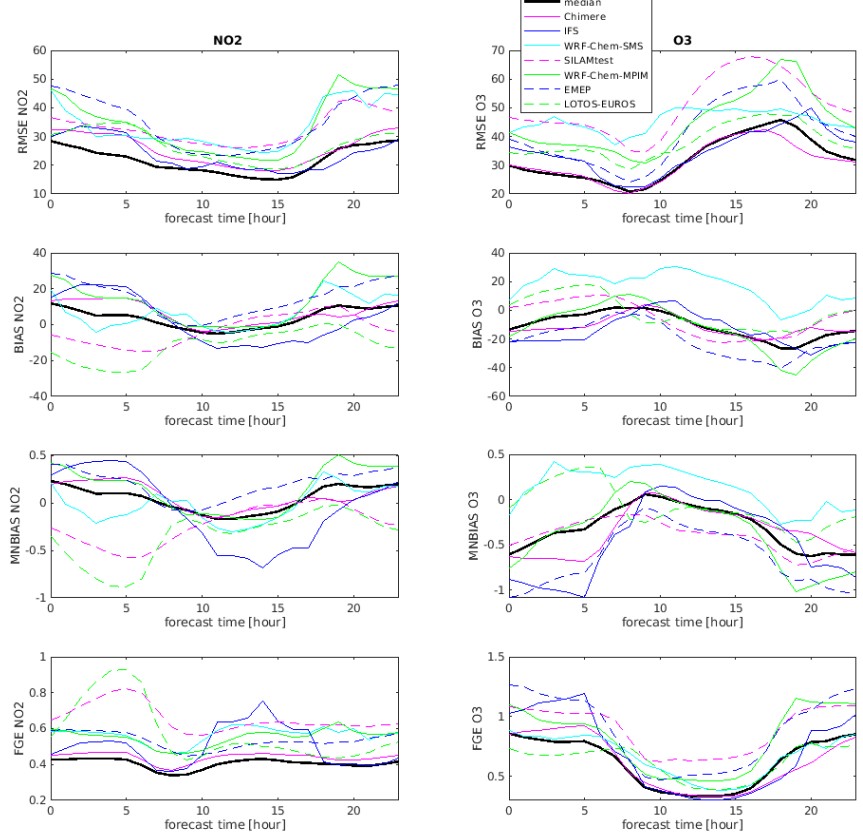

*Figure 6: RMSE, BIAS, MNBIAS and FGE of NO₂ and O₃ over the forecasting time (time of the*
*day).*






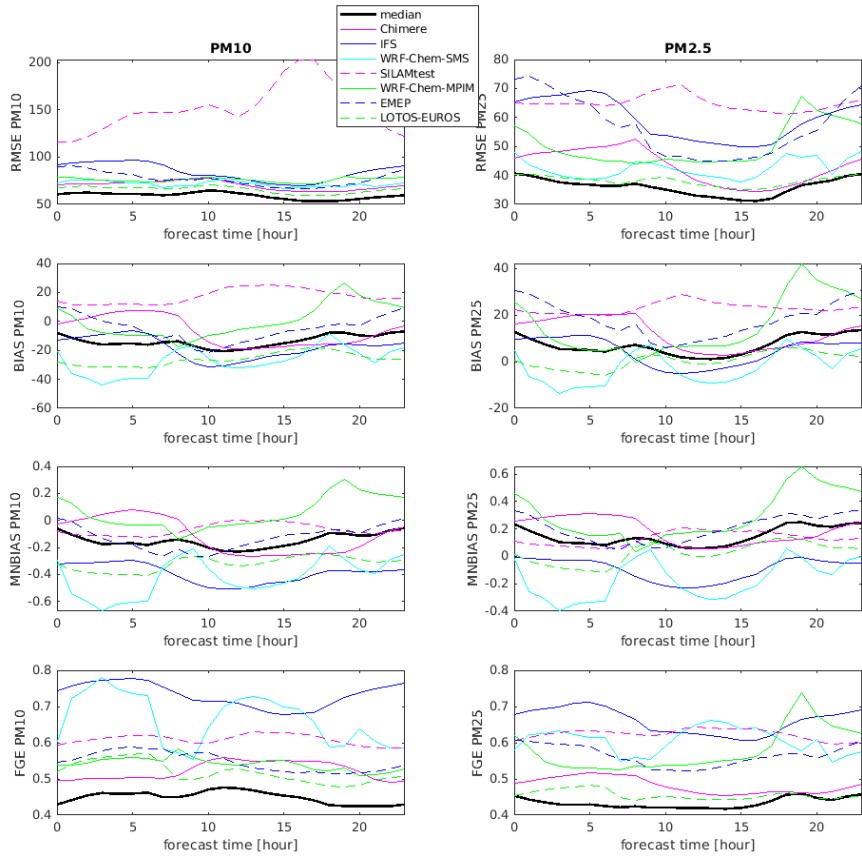

*Figure 7: RMSE, BIAS, MNBIAS and FGE of PM10 and PM2.5 over the forecasting time (time of*
*the day).*






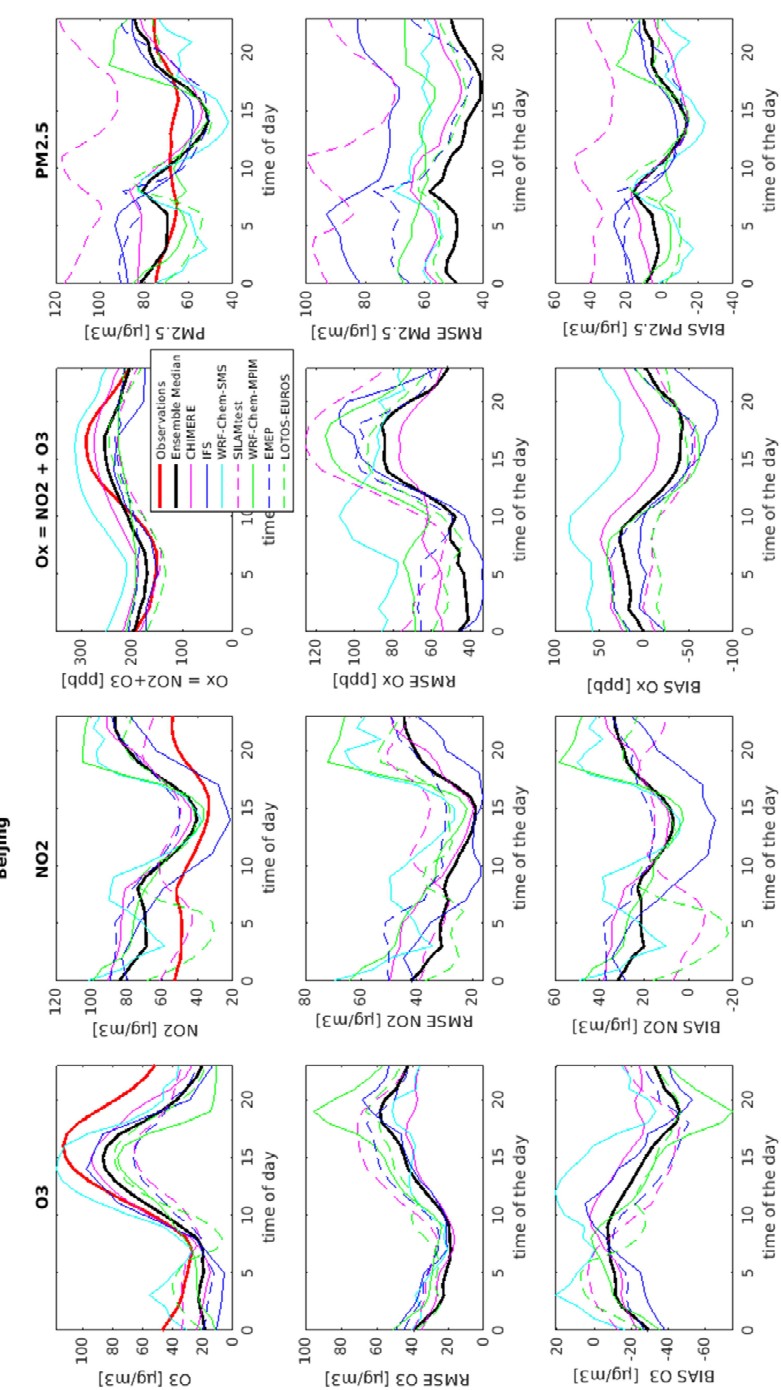

*Figure 8: Diurnal variations of the concentrations and of the RMSE and BIAS of O₃, NO₂, Oₓ and*
*PM2.5 for Beijing for the whole time period (April 2016 – June 2017).*






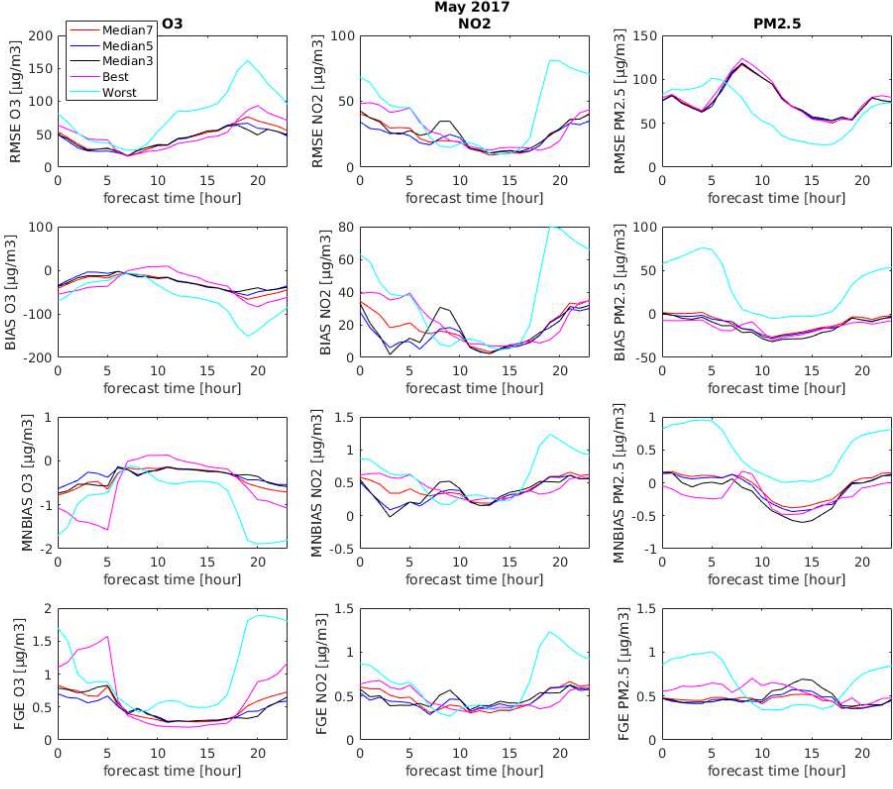

*Figure 9: RMSE, BIAS, MNBIAS and FGE of O₃, NO₂ and PM2.5 over the forecasting time (time of*
*the day) for the Median7, Median5, Median3 and the best and the worst model.*





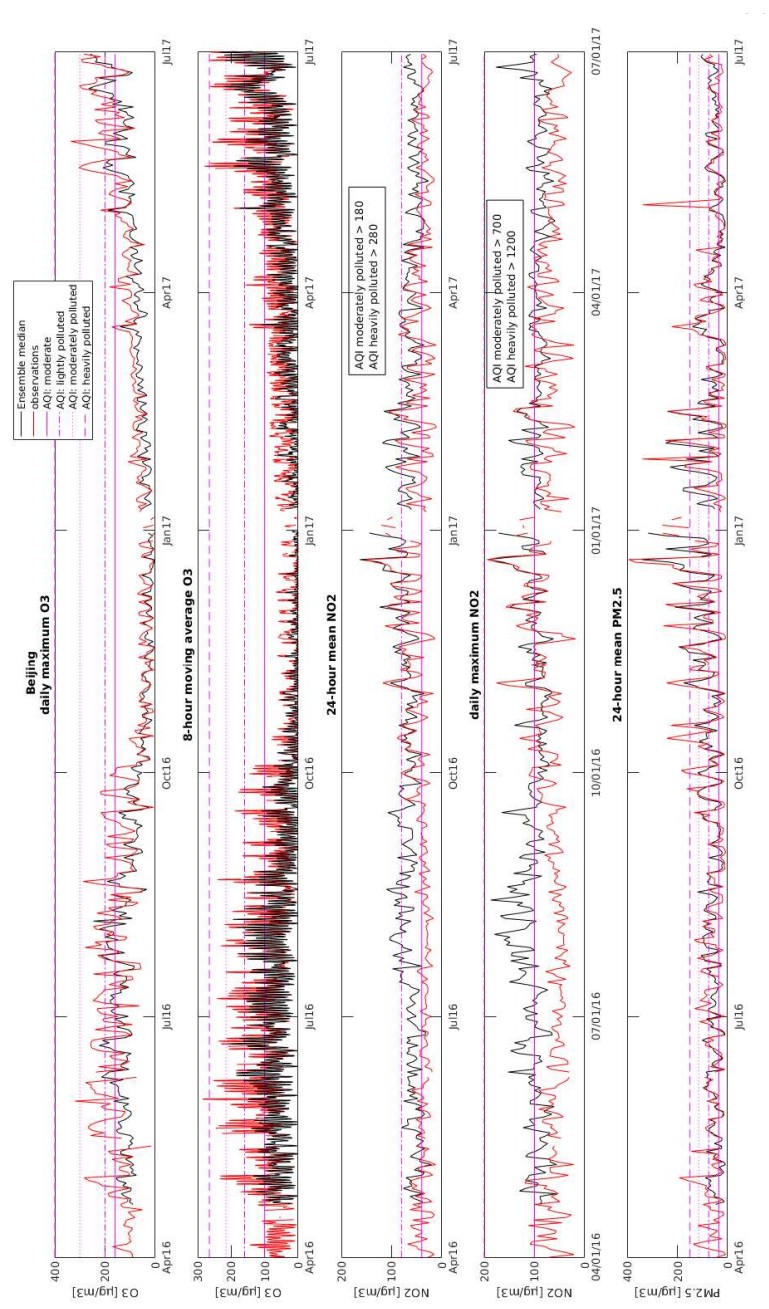

996

997 *Figure 10: Timeseries of daily maximum O₃, 8-hour moving average O₃, 24-hour mean NO₂, daily*
998 *maximum NO₂ and 24-hour mean PM2.5 for Beijing from April 2016 until June 2017.*





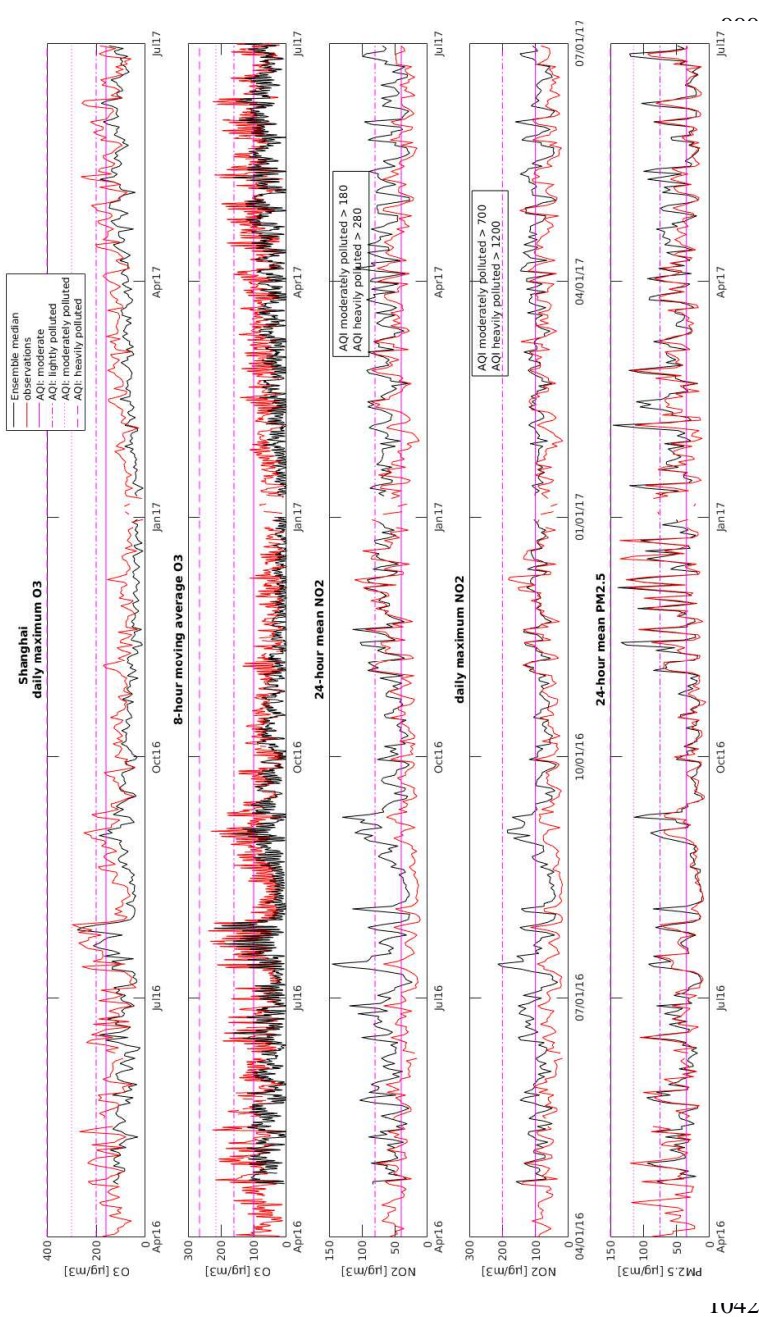

Figure 11: Timeseries of daily maximum O₃, 8-hour moving average O₃, 24-hour mean NO₂, daily maximum NO₂ and 24-hour mean PM2.5 for Shanghai from April 2016 until June 2017.





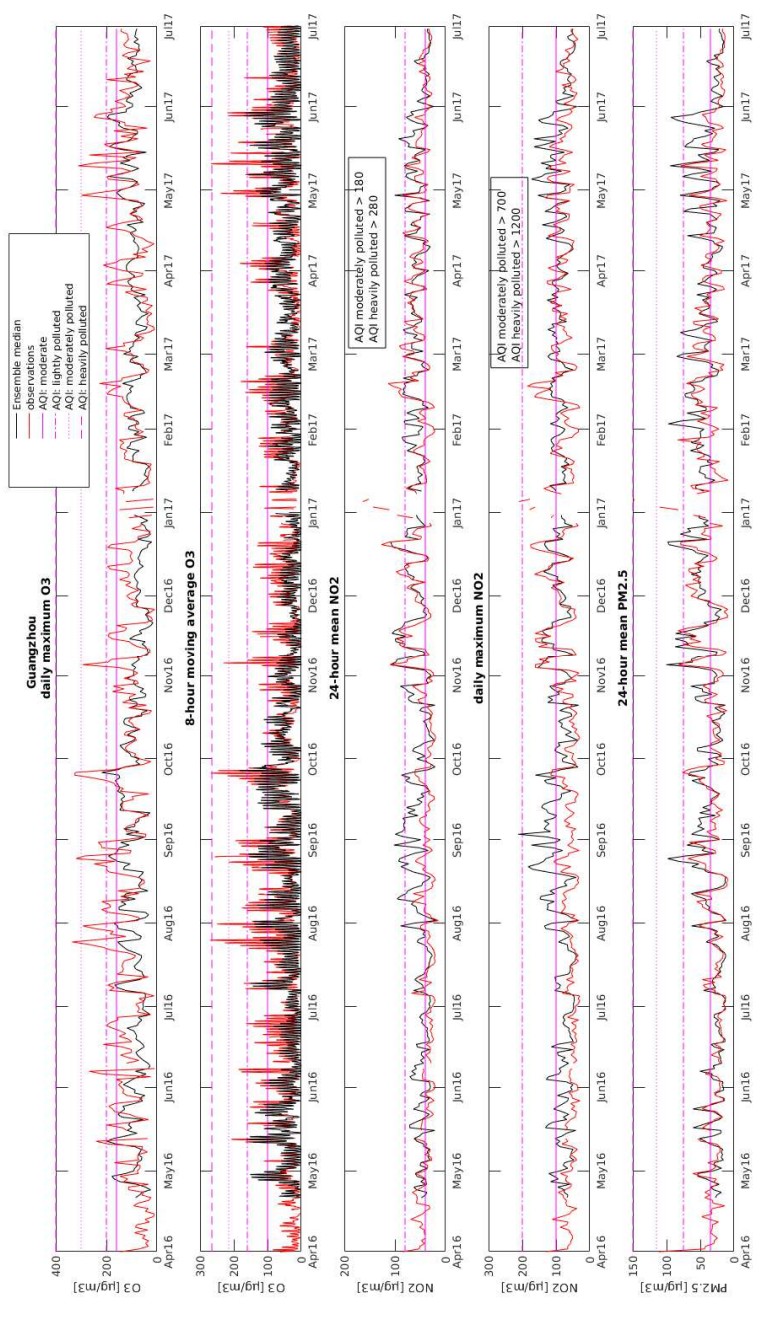

*Figure 12: Calculated (ensemble median) and observed timeseries of daily maximum O₃, 8-hour*
*moving average O₃, 24-hour mean NO₂, daily maximum NO₂ and 24-hour mean PM2.5 for*
*Guangzhou from April 2016 until June 2017.*





*Figure 13 a and b: Timeseries of calculated (ensemble*
*median) and observed daily maximum and 8-hour moving average O₃ for Beijing and Shanghai*
*together with the bias corrected calculated timeseries.*





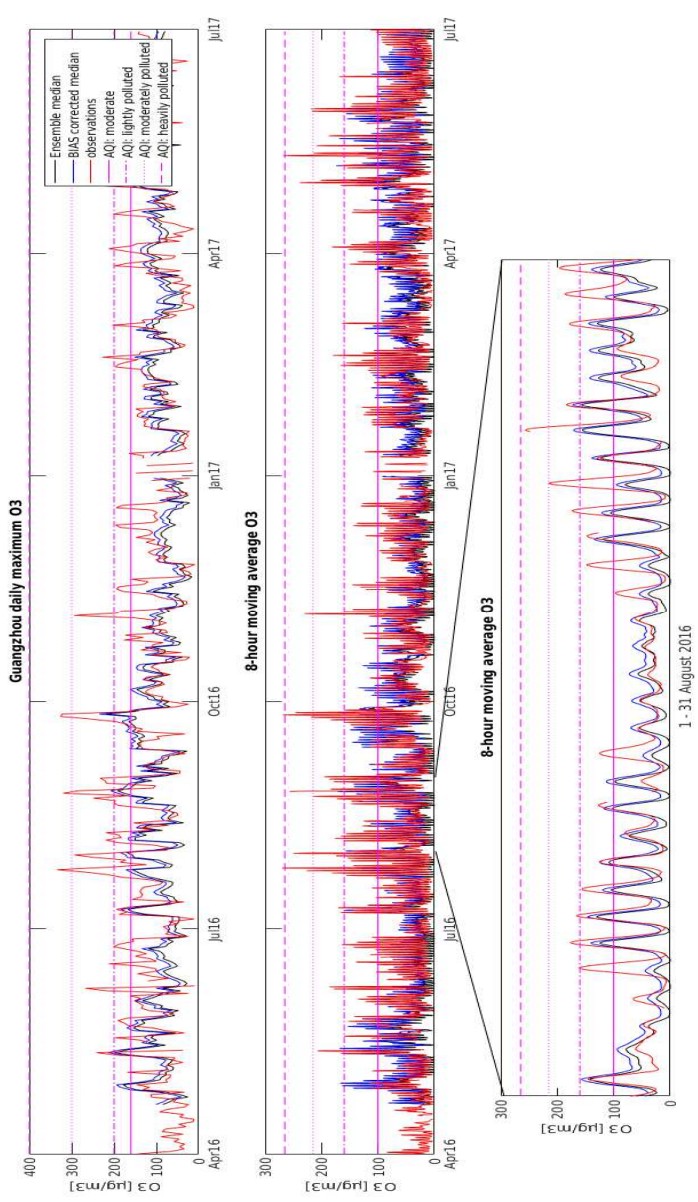


*Figure 13 c: Timeseries of calculated (ensemble median) and observed daily maximum and 8-hour*
*moving average O₃ for Guangzhou together with the bias corrected calculated timeseries.*

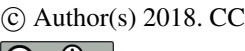



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
