# Peer review of "Ensemble Forecasts of Air Quality in Eastern China 1"

_Geoscientific Model Development, 2018_

## Referee Comment (RC1) · Anonymous Referee #1 · 13 Oct 2018

General comments

This paper presents an operational multi-model forecasting system for air quality which has been developed to provide air quality services for urban areas of China (MarcoPolo-Panda). A companion paper describes the models involved in the fore-casting platform. The prediction system provides a 72 hours forecast with a zoom over the largest Chinese cities. In the present paper the performances of each member and the ensemble are evaluated with available data for the main criteria pollutants. This platform will be useful to develop services and improved in a near future by adding new members of the ensemble, use of satellite data for data assimilation and other MOS

technique to improve the forecasting system. I am favorable to the publication of this work.

The author should add the sulfur dioxide in the analysis since this pollutant is in the target of the Chinese authorities in the new 3 years plan adopted to reduce air pollution in China. Even if the model description is provided in the companion paper, a short description of models with the main features like resolutions, meteorological and emission data must be provided. As it is an analysis based on the first 24hours forecast, the ability of models to predict air pollutant concentrations could be due to the ability of the met driver to forecast the meteorological conditions or to the chemistry transport model itself.

Specific comments

As it is written and specified in the paper the main weakness of the exercise is to run models with outdated emissions. This must be improved in the next versions of the platform with certainly a yearly update. To better understand the behavior on PM25 concentrations a short analysis on sulfate, ammonium and nitrate species could be added to highlight the main pattern of these species. The authors must justify why they did not include peri urban and rural stations in the vicinity of cities to better match with the resolution of models. I suggest the authors to remind how POD, FAR and AQI indexes are calculated or they must add a reference. L. 283: These low correlations for PM could be due to dust storms occurring during this period, Usually, for pollutants the occurrence of pollution events increase the time correlation that's why ozone time correlations are better in summer and PM correlation better in winter. To understand the negative bias on PM10, the authors should remind how dust are taken into account by models (boundary conditions, local emission parametrizations). The author should comment the flat diurnal profile of PM2.5 in the observations while the model are very sensitive to the increase of the PBL during the afternoon, perhaps the effect of the secondary production of aerosols that is not well predicted by the models. Boundary conditions are important drivers for several pollutant concentrations such as Ozone,

[Figure]

PM10, PM2.5, dust, sulfates. This should be highlighted in this study. In this paper or in the companion paper an analysis of the behavior of the model used for the boundary conditions would be helpful to understand the performances of the models. I suggest to cite this paper : Bessagnet, B., Pirovano, G., Mircea, M., Cuvelier, C., Aulinger, A., Calori, G., Ciarelli, G., Manders, A., Stern, R., Tsyro, S., García Vivanco, M., Thunis, P., Pay, M.-T., Colette, A., Couvidat, F., Meleux, F., Rouïl, L., Ung, A., Aksoyoglu, S., Baldasano, J. M., Bieser, J., Briganti, G., Cappelletti, A., D'Isidoro, M., Finardi, S., Kranenburg, R., Silibello, C., Carnevale, C., Aas, W., Dupont, J.-C., Fagerli, H., Gonzalez, L., Menut, L., Prévôt, A. S. H., Roberts, P., and White, L.: Presentation of the EU-RODELTA III intercomparison exercise – evaluation of the chemistry transport models' performance on criteria pollutants and joint analysis with meteorology, Atmos. Chem. Phys., 16, 12667-12701, https://doi.org/10.5194/acp-16-12667-2016, 2016. This paper provides an intercomparison of some of the models used in the MarcoPolo-Panda project. L. 106 : "We show that the application of bias correction to the models improves the forecasting skills of binary ozone predictions". This sentence cannot be written in the introduction as it is a concluding remark. L. 306 : For the overestimation of NO2 over cities where emissions are better documented, the missing urban parameterization could be one of the reason due to less vertical mixing in the model. L. 647 Is it a correction based on analysis or forecast? The method is not well described. L.756 "...predicts the occurrence of pollution events a few days before they occur." Difficult to write this as the author only focus on the first 24 hours forecast. L.780 "Furthermore, data assimilation of satellite and in situ observations should significantly improve the performance of the forecasting system." This is very challenging but promising, the authors could add some references to support this initiative in terms of added value for such a system. L782 What the authors mean by "a more advanced approach"? I think that MOS (Model Output Statistics) techniques applied to the ENSEMBLE could be more useful to improve such a forecast system.

Technical comments

All along the paper put ensemble in capital letter ENSEMBLE Table 3: Replace NaN by NA (not applicable) since NaN means Not a number Figures 5,6,7 : Units? Certainly the quality of figure can be improved and the figures harmonized.

L. 166 by relatively coarse resolution models L.225 Should be ...“nitrogen” emissions L.279 ...”exhibit small correlation”..., I would say "low" correlations L313-314 “These cities also show an overestimation of $NO_2$ concentrations” would be better L.612 The predictions of PM2.5 concentrations

Please also note the supplement to this comment:
https://www.geosci-model-dev-discuss.net/gmd-2018-234/gmd-2018-234-RC1-supplement.pdf

---

## Referee Comment (RC2) · Anonymous Referee #2 · 10 Dec 2018

This paper is the second of the two papers that describes a multi-model ensemble air quality forecasting system developed by a consortium of European and Chinese scientists under two EU funded projects Marcopolo and PANDA. This paper presents a detailed evaluation of the forecasting system for just over an year. The analysis is comprehensive and the results are presented very well. The authors clearly demonstrate the value of ensemble forecasts and unsurprisingly shows that ensemble forecast has more skill compared to individual models. I recommend publication of the paper after following minor revisions.

Figure 5: Suggest adding more labels in BIAS colorbar and specifying unite for BIAS

[Figure]

and RMSE.

Line 399: change that to than

Lines 430-437: Are these criteria adopted in the operational system?

Line 491: Change indexes to indices.

---

## Author Comment (AC1) · 7 Jan 2019

Author's Comments to Reviewer Comments:

Reply to Review1:

General comments of RW1: This paper presents an operational multi-model forecasting system for air quality which has been developed to provide air quality services for urban areas of China (MarcoPolo-Panda). A companion paper describes the models involved in the forecasting platform. The prediction system provides a 72 hours forecast with a zoom over the largest Chinese cities. In the present paper the performances of

each member and the ensemble are evaluated with available data for the main criteria pollutants. This platform will be useful to develop services and improved in a near future by adding new members of the ensemble, use of satellite data for data assimilation and other MOS technique to improve the forecasting system. I am favorable to the publication of this work.

The author should add the sulfur dioxide in the analysis since this pollutant is in the tar- get of the Chinese authorities in the new 3 years plan adopted to reduce air pollution in China. Even if the model description is provided in the companion paper, a short description of models with the main features like resolutions, meteorological and emission data must be provided.

Author comment: Sulfur dioxide is indeed an important component of the air pollution problem since it is directly emitted as a result of coal burning. Unfortunately, the MarcoPolo/Panda website does not keep track of the forecasts made by some of the models nor does it record the observational data. This should be addressed in the updated version of the forecast system and of the web site maintained by our colleagues at KNMI in the Netherlands.

Changes in revised manuscript: A table is added that recap the main features of the models involved.

Review Comment: As it is an analysis based on the first 24hours forecast, the ability of models to predict air pollutant concentrations could be due to the ability of the met driver to forecast the meteorological conditions or to the chemistry transport model itself.

Author comment: Yes, the major drivers for the prediction over 24 hours are (1) the initial conditions, (2) the meteorological forecast and (3) to a lesser extent the emissions. The behavior of the boundary layer also plays an important role. This point is now highlighted in the paper.

[Figure]

Specific comments Reviewer1:

As it is written and specified in the paper the main weakness of the exercise is to run models with outdated emissions. This must be improved in the next versions of the platform with certainly a yearly update. To better understand the behavior on PM25 concentrations a short analysis on sulfate, ammonium and nitrate species could beadded to highlight the main pattern of these species. The authors must justify why they did not include peri urban and rural stations in the vicinity of cities to better match with the resolution of models.

Author comment: The MarcoPolo/Panda project was developed at a time where only the observations of a few urban sites were available. It was very unclear if, during the execution of the project, some data would be made available. Fortunately, the situation in China changed in the meantime. Nevertheless, the project accepted by the EU had an emphasis on urban air pollution. If funding is made available, we would very much like to add rural sites in the analysis and focus more on ammonium and nitrate species. We would like also to put more focus on deposition issues, specifically in agricultural areas.

Review Comment: I suggest the authors to remind how POD, FAR and AQI indexes are calculated or they must add a reference.

Author Comment: POD and FAR are shortly explained in the text. Now, the reference Brasseur et al. has been added again for clarification. AQI is not calculated from our data. We are using the thresholds of the Chinese Government (see the tables). We calculate from our 1-hourly time series the time series for 1) 1-hour ozone, 2) 8-hour ozone concentrations 3) 24-hour mean NO2 concentrations, 4) 1-hour NO2 concentrations and 5) 24-hour PM2.5 concentrations to apply the AQI as they are defined (e.g. for 8-hour ozone). For clarification, we will change the sentence accordingly: "The air quality indices are calculated for 1) 1-hour ozone, 2) 8-hour ozone concentrations 3) 24-hour mean NO2 concentrations, 4) 1-hour NO2 concentrations and 5) 24-hour
PM2.5 concentrations." is changed to: "Based on the 1-hourly time series of ozone, NO2 and PM2.5, the time series for 1) 1-hour ozone, 2) 8-hour ozone concentrations 3) 24-hour mean NO2 concentrations, 4) 1-hour NO2 concentrations and 5) 24-hour PM2.5 concentrations have been constructed and the thresholds of the air quality indices (AQI) have been applied for each definition."

The reference for POD and FAR calculation (Brasseur et al.) is added to the revised manuscript.

Review Comment: L. 283: These low correlations for PM could be due to dust storms occurring during this period, Usually, for pollutants he occurrence of pollution events increase the time correlation that's why ozone time correlations are better in summer and PM correlation better in winter. To understand the negative bias on PM10, the authors should remind how dust are taken into account by models (boundary conditions, local emission parametrizations).

Author Comment: Each model is using its own approach to calculate the mobilization of the dust, specifically in the deserts. Table 3 of the companion paper by Brasseur et al. provides information about the dust emission used by the different models. As you will note from this Table, some models do not include dust mobilization.

Revised Manuscript: We have added the following text at the end of Section 3.1 just after the words "...model tunings" and before the words "For the entire time period...." The text is: "An important difference between the models included in the ensemble is the formulation of dust mobilization (see Table 3 of the companion paper by Brasseur et al., 2018). Note that the CHIMERE and EMEP models do not include dust in their calculation of particulate matter and that the emissions provided by the IFS-ECMWF are substantially higher than in other models."

Review Comment: The author should comment on the flat diurnal profile of PM2.5 in the observations while the model are very sensitive to the increase of the PBL during the afternoon, perhaps the effect of the secondary production of aerosols that is not

well predicted by the models. Boundary conditions are important drivers for several pollutant concentrations such as Ozone, PM10, PM2.5, dust, sulfates. This should be highlighted in this study.

Author Comment: The diurnal variation of PM2.5 is very similar to that of NO2. These two compounds are to a large extent released at the surface in the boundary layer and are in part ventilated to the free troposphere during day by convective motion and mixing. During the night, the boundary layer becomes shallow and stable, and the convective motions are therefore interrupted. As a result, one expects an increase in the surface concentration of the species emitted at the surface including NO2 and PM2.5. The models tend to confirm this view. However, the observational data do not show with such a large amplitude the day/night difference that the models simulate. The reason for this discrepancy is not fully understood, and we are currently working on this question. It seems that in urban areas, the heat produced by the buildings and other human activities as well as the turbulence generated by the urban canopy is sufficient to produce some turbulent mixing and ventilation of species. These urban effects, that would tend to make the diurnal evolution more flat are not well reproduced by the current models and this is a question for which some improvement can be made in the future.

Revised Manuscript: We are adding at the end of Section 3 after the words PBL the following text: "Specifically, one should note that the models do not include a detailed formulation if small scale urban canopy effects, which could generate some mechanic and thermal turbulence with related vertical mixing during nighttime. With increased nighttime ventilation from the boundary layer to the free troposphere, the calculated amplitude of the diurnal variation of gases and particulates would be reduced and become closer to the observation."

Review Comment: In this paper or in the companion paper an analysis of the behavior of the model used for the boundary conditions would be helpful to understand the performances of the models. I suggest to cite this paper : Bessagnet, B., Pirovano,

G., Mircea, M., Cuvelier, C., Aulinger, A.,Calori, G., Ciarelli, G., Manders, A., Stern, R., Tsyro, S., García Vivanco, M., Thunis, P., Pay, M.-T., Colette, A., Couvidat, F., Meleux, F., Rouïl, L., Ung, A., Aksoyoglu, S., Baldasano, J. M., Bieser, J., Briganti, G., Cappelletti, A., D'Isidoro, M., Finardi, S., Kranenburg, R., Silibello, C., Carnevale, C., Aas, W., Dupont, J.-C., Fagerli, H., Gonzalez, L., Menut, L., Prévôt, A. S. H., Roberts, P., and White, L.: Presentation of the EU-RODELTA III intercomparison exercise – evaluation of the chemistry transport models'performance on criteria pollutants and joint analysis with meteorology, Atmos. Chem. Phys., 16, 12667-12701, https://doi.org/10.5194/acp-16-12667-2016, 2016. This paper provides an intercomparison of some of the models used in the MarcoPolo-Panda project.

Author Comment: Thank you for the comment. Revised Manuscript: The citation is added to the revised manuscript. We have added a sentence after the second paragraph of Section 2. "Several of the models considered here have been involved in a previous intercomparison summarized by Bessagnet et al. (2016)."

Review Comment: L. 106 : "We show that the application of bias correction to the models improves the forecasting skills of binary ozone predictions". This sentence cannot be written in the introduction as it is a concluding remark.

Author Comment: Sentence is be removed in the revised paper.

Review Comment: L. 306 : For the overestimation of NO2 over cities where emissions are better documented, the missing urban parameterization could be one of the reason due to less vertical mixing in the model.

Author comment: We have added a sentence to specify this in the revised manuscript.

Review Comment: L. 647 Is it a correction based on analysis or forecast? The method is not well described.

Author comment: The correction is based on the 0-24h forecast (the data, which is saved in the system and the only data available for this evaluation).

Review Comment: L.756 ". . .predicts the occurrence of pollution events a few days before they occur." Difficult to write this as the author only focus on the first 24 hours forecast.

Author comment: The prediction system provides every day the forecast of the next 3 days. Unfortunately, only the data of the 0-24h forecast can be saved and is available for this evaluation. But the prediction of the next three days is available every day to the public.

Review Comment: L.780 "Furthermore, data assimilation of satellite and in situ observations should significantly improve the performance of the forecasting system." This is very challenging but promising, the authors could add some references to support this initiative in terms of added value for such a system.

Author's response: We have added in the text after this sentence the words: "(see e.g., Mizzi et al., 2016)"

Mizzi, A.P., A.F. Arellano, D.P. Edwards, J.L. Anderson, and G.G. Pfister: Ăă Assimilating compact phase space retrievals of atmospheric composition with WRF-Chem/DART: a regional chemical transport/ensemble Kalman filter data assimilation system, Geosci. Model Dev.,Ăă9, 965-978, 2016.

Review Comment: L782 What the authors mean by "a more advanced approach"? I think that MOS (Model Output Statistics) techniques applied to the ENSEMBLE could be more useful to improve such a forecast system.

Author Comment: We thought e.g. about an approach taking the seasonal and also the daily variation into account (a more correct diurnal cycle), this is part of MOS.

Technical Comments Review1:

All along the paper put ensemble in capital letter ENSEMBLE

Author comment: We would prefer to keep "ensemble" as it is, because we do not use

it often similar to a model name, and often use "ensemble median" or "ensemble of models". In addition, in the accepted companion paper, we use ensemble instead of ENSEMBLE, and we would prefer to keep it homogeneous. If the reviewer accepts our preference, we would like to keep it.

Review Comment: Table 3: Replace NaN by NA (not applicable) since NaN means Not a number

Okay, done in the revised paper

Review Comment: Figures 5,6,7 : Units? Certainly the quality of figure can be improved and the figures harmonized.

Units have been added to the figure, and quality is improved, figures harmonized.

Review Comments: L. 166 by relatively coarse resolution models

Done

L.225 Should be . . ."nitrogen" emissions

Done

L.279 . . ."exhibit small correlation". . ., I would say "low" correlations

Done

L313-314 "These cities also show an overestimation of NO2 concentrations" would be better

Done

L.612 The predictions of PM2.5 concentrations

Done

Reply to Review2:

General comments of RW2:

This paper is the second of the two papers that describes a multi-model ensemble air quality forecasting system developed by a consortium of European and Chinese scientists under two EU funded projects Marcopolo and PANDA. This paper presents adetailed evaluation of the forecasting system for just over an year. The analysis is comprehensive and the results are presented very well. The authors clearly demonstrate the value of ensemble forecasts and unsurprisingly shows that ensemble forecast has more skill compared to individual models. I recommend publication of the paper after following minor revisions.

Figure 5: Suggest adding more labels in BIAS colorbar and specifying unite for BIAS

Changed in the revised manuscript.

Line 399: change that to than

Done, thank you!

Lines 430-437: Are these criteria adopted in the operational system?

Author comment: In the operational system, the ensemble median is calculated based on the median of all available models for each hour. If one (or more) model is occasionally missing, the ensemble median is calculated based on the median of the remaining models. The criteria MEDIAN3, MEDIAN5, etc. have been calculated for testing the performance of the system, but only for the test period.

Line 491: Change indexes to indices. Done

Please also note the supplement to this comment:
https://www.geosci-model-dev-discuss.net/gmd-2018-234/gmd-2018-234-AC1-supplement.pdf

---

## Author Comment (AC2) · 7 Jan 2019

[revised manuscript text omitted]

(15/94) | 0.03
(1/36) | 0
(0/5) | 0.21
(4/19) | 0
(0/1) | NA
(0/0) |
| **Bias corrected 1-hour O₃ [µg m⁻³]** | 0.32
(30/94) | 0.14
(5/36) | 0
(0/5) | 0.33
(15/45) | 0.29
(2/7) | NA
(0/0) |
| **8-hour O₃ [µg m⁻³]** | 0.31
(315/1032) | 0.06
(12/217) | 0
(0/47) | 0.28
(122/437) | 0
(0/12) | NA
(0/0) |
| **Bias corrected 8-hour O₃ [µg m⁻³]** | 0.49
(508/1032) | 0.13
(29/217) | 0
(0/47) | 0.37
(296/804) | 0.19
(7/36) | NA
(0/0) |
| **24-hour NO₂ [µg m⁻³]** | 0.94
(208/222) | 0.56
(15/27) | NA
(0/0) | 0.35
(110/318) | 0.68
(32/47) | NA
(0/0) |
| **1-hour NO₂ [µg m⁻³]** | 0.76
(58/76) | NA
(0/0) | NA
(0/0) | 0.63
(97/155) | 1
(1/1) | NA
(0/0) |
| **24-hour PM2.5 [µg m⁻³]** | 0.85
(149/175) | 0.57
(4/7) | NA
(0/0) | 0.30
(65/214) | 0.80
(16/20) | NA
(0/0) |

[revised manuscript text omitted]

13852387-2404, doi:10.5194/acp-15-2387-2015, 2015.